# GraphGT: Machine Learning Datasets for Graph Generation and Transformation

**Yuanqi Du**[1]*, **Shiyu Wang**[2]*, **Xiaojie Guo**[5], **Hengning Cao**[1], **Shujie Hu**[2], **Junji Jiang**[3],
**Aishwarya Varala**[1], **Abhinav Angirekula**[4], **Liang Zhao**[2]†

[1]George Mason University, [2]Emory University, [3]Tianjin University, [4] Thomas Jefferson High School,
[5]JD.COM Silicon Valley Research Center
contact: `ydu6@gmu.edu`, `liang.zhao@emory.edu`

## Abstract

Graph generation has shown great potential in applications like network design and mobility synthesis and is one of the fastest-growing domains in machine learning for graphs. Despite the success of graph generation, the corresponding real-world datasets are few and limited to areas such as molecules and citation networks. To fill the gap, we introduce GraphGT, a large dataset collection for graph generation and transformation problem, which contains 36 datasets from 9 domains across 6 subjects. To assist the researchers with better explorations of the datasets, we provide a systemic review and classification of the datasets based on research tasks, graph types, and application domains. We have significantly (re)processed all the data from different domains to fit the unified framework of graph generation and transformation problems. In addition, GraphGT provides an easy-to-use graph generation pipeline that simplifies the process for graph data loading, experimental setup and model evaluation. Finally, we compare the performance of popular graph generative models in 16 graph generation and 17 graph transformation datasets, showing the great power of GraphGT in differentiating and evaluating model capabilities and drawbacks. GraphGT has been regularly updated and welcomes inputs from the community. GraphGT is publicly available at `https://graphgt.github.io/` and can also be accessed via an open Python library.

## 1   Introduction

Graphs are ubiquitous data structures to capture connections (i.e., edges) between individual units (i.e., nodes). One central problem in machine learning on graphs is the gap between the discrete graph topological information and continuous numerical vectors preferred by data mining and machine learning models [1, 2, 3]. This directly leads to two major directions on graph research in modern machine learning: 1) graph representation learning [2, 4, 5, 6], which aims at encoding graph structural information into a (low-dimensional) vector space, and 2) graph generation [7, 8], which reversely aims at constructing a graph-structured data from the (low-dimensional) vector space. In the past several years, graph representation learning has enjoyed an explosive growth in machine learning. Techniques such as DeepWalk [9], graph convolutional network (GCN) [10], and graph attention networks (GAT) [11] have been proposed for various tasks including node classification [12], link prediction [13, 14, 15], clustering [2, 4] and others [16, 17].

Beyond graph representation learning, graph generation and transformation via machine learning start to obtain fast-increasing attention in even more recent years. It enables end-to-end learning of underlying unknown graph generation or transformation process, which is a significant advancement beyond traditional prescribed graph models such as random graphs and stochastic block models

---

*Equal Contribution
†Corresponding Author

Submitted to the 35th Conference on Neural Information Processing Systems (NeurIPS 2021) Track on Datasets and Benchmarks. Do not distribute.

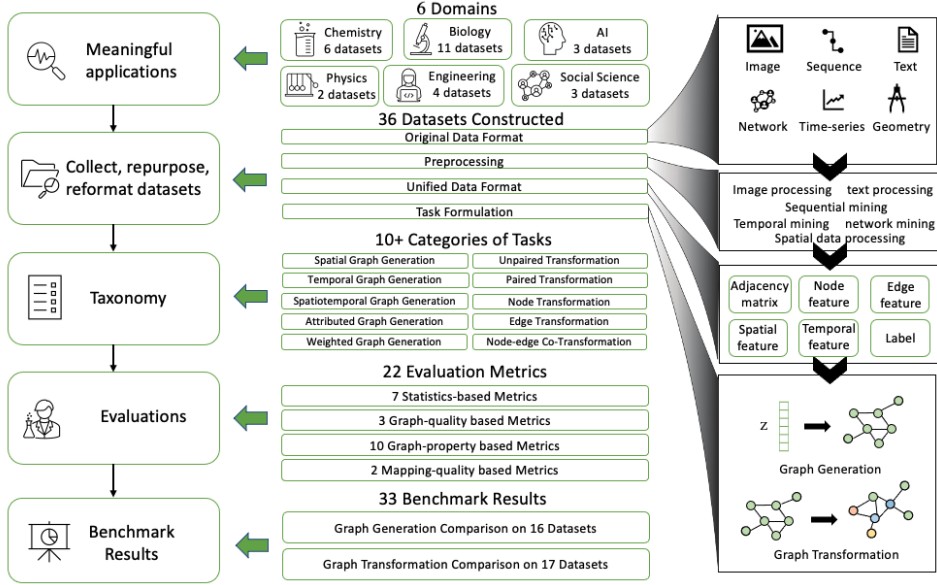

**Figure 1:** GraphGT dataset collection overview.

which require strong human prior knowledge and hand-crafted rules. Hence, graph generation and transformation via machine learning has great potential of many challenging tasks such as molecule design, mobility network synthesis, and protein folding statistical modeling. Over recent few years, substantial efforts have been paid on developing models and algorithms for graph generation and transformation, and a few of them have been studied targeting specific domains, such as GraphVAE [18], MolGAN [19] and JT-VAE [20].

Unlike graph representation learning which enjoys various benchmark datasets such as CORA, CITESEER and PUBMED for node classification [21], OAG for link prediction [22], and Molecule-LENET for graph-level prediction [23], graph generation and transformation via machine learning is still in its nascent stage and lacks comprehensive benchmark datasets that well cover different key real-world applications and types of graph patterns. Existing datasets are basically limited to few domains such as citation networks and molecules [7, 24]. Such data scarcity issue further leads to the following bottlenecks for the advancement of this fast-growing domain of graph generation and transformation: **(1) Difficulty in formulation:** graph structured data is complex in its nature; and the raw data in different domains may requires greatly different procedures to process or re-process in order to fit into a unified format. **(2) Limited number of application domains:** Although graph generation and transformation is a very broad generic concept that covers graphs in areas ranging from geography to biology, to physics, to sociology, to engineering, existing datasets only cover limited domains which prevents the development of graph generative models as well as applications in more diverse domains. **(3) Lack of taxonomy:** As the area of graph generation and transformation grows, the research tasks are diversified and hence require a well-defined categorization in order to have the dataset under the right category for the evaluation of the corresponding task. **(4) Lack of unified evaluation procedures:** the evaluation metrics used in existing research works are quite diverse and a gold standard for the evaluation procedure and metrics is needed. Moreover, the scarcity of existing datasets may bias the selection of elevation metrics to fit the limited number of existing datasets (e.g., molecules) but may not be general to other datasets. **(5) Lack of comprehensive model comparisons:** existing models are usually evaluated in a small number of datasets in very focused domains and some may be prone to "overfitting" to these datasets already, which significantly challenges the differentiation, evaluation, and advancement of the existing methods.

To tackle the aforementioned challenges, we introduce GraphGT, a large dataset collection for graph generation and transforamtion in machine learning, which **(1)** collects, re-purposes, re-formats a large amount of graph datasets, that **(2)** covers a variety of domains and subjects, **(3)** provides a systematic reviews and classifications of the datasets, **(4)** standardizes on the model evaluation procedures, and **(5)** provides benchmark results on a large amount of datasets. The major contributions are as follows.

- 36 datasets are published under various graph types cover 6 disciplines (including biology, physics, chemistry, artificial intelligence (AI), engineering, and social science) and 9 domains (including

protein, brain network, physical simulation, vision, molecule, transportation science, electrical and computer engineering (ECE), social network and synthetic data).

- Among all 36 datasets, we collect and construct CollabNet dataset and 7 brain network datasets from scratch for graph generation and transformation. Another 8 datasets are re-purposed by us from other applications into graph generation and transformation tasks for the first time. The remaining are from very different domains that share quite different terminology, formats, and data structures, which are reformatted by us to a unified format for the first time for easy access and use in a standardized manner.

- We provide and analyze results of graph generation on 16 datasets and graph transformation on 17 datasets using popular graph generation and transformation models. We observed that the performance of the comparison methods in different datasets (e.g., with different graph sizes, feature types, etc.) in different domains can be quite diverse. Hence GraphGT can be very helpful in differentiating the comparison methods, locating their drawbacks, and further advancing them.

- Easy-to-use Python API for users to query and access pre-processed datasets according to specific disciplines, domains, and applications per their interests. We also provide a detailed tutorial for the implementation in Appendix E. In addition to the access via the Python API, GraphGT is open-sourced and available for download via GitHub at https://graphgt.github.io/.

## 2 Related Works

As graph representation learning enjoys an explosive growth in machine learning, numerous research works such as DeepWalk [9], graph convolutional network (GCN) [10], and graph attention networks (GAT) [11] have been proposed for various tasks including node classification [12], link prediction [13, 14] and clustering [2, 4]. Along with this, some datasets are proposed, such as datasets for node classification (CORA, CITESEER and PUBMED) [21], datasets for link prediction (OAG) [22], and datasets for Graph-level prediction (Molecule-LENET) [25]. To summarize and standardize these datasets, many data collections for graph representation learning has been proposed. Stanford Network Analysis Platform (SNAP) is a general purpose network analysis and graph mining library which contains massive networks with hundreds of millions of nodes, and billions of edges [26]. OPEN GRAPH BENCHMARK (OGB) is a diverse set of challenging and realistic benchmark datasets to facilitate scalable, robust, and reproducible graph machine learning (ML) research [27]. However, most of the datasets for graph representation learning research cannot be used as graph generation benchmarks as the latter requires large number of individual whole graphs in order to learn their distributions. While the aforementioned datasets either contain one giant graph for node classification and link prediction, or a set of graphs from different distributions for graph classification.

Graph generation and transformation have been increasingly drawing attentions from the community due to its significant roles in various domains. Though many advanced methods have been proposed, there are only limited number of datasets for this research topics. Enzyme dataset [28], ProFold dataset [29] and Protein dataset [30] are used for protein structure generation. ZINC molecule database is borrowed to generate optimal molecules that have desired properties [20]. Moreover, a few synthetic datasets are also generated for graph generation tasks to learn graph distributions, such as Erdos-Renyi graphs [31] and Waxman random graphs [29]. There exist few data collections that systematic organize the graph generation datasets from different domains.

## 3 Graph Generation and Transformation

A graph can be defined as $G = (\mathcal{V}, \mathcal{E}, E, F)$, where $\mathcal{V}$ is the set of $N$ nodes, and $\mathcal{E} \subseteq \mathcal{V} \times \mathcal{V}$ corresponds to a set of edges. $e_{ij} \in \mathcal{E}$ is an edge that connects node $v_i$ and $v_j \in \mathcal{V}$. If the graph is node-attributed or edge-attributed, it has the node attribute matrix $F \in \mathbb{R}^{N \times D}$ that assigns node attributes to each node or edge attribute tensor $E \in \mathbb{R}^{N \times N \times K}$ that assigns attributes to each edge. $D$ and $K$ are dimensions of node attributes and edge attributes, respectively.

### 3.1 Graph Generation

Graph generation aims to sample novel graphs via well-designed probabilistic models [7]. More formally, given a set of observed graphs with arbitrary number of nodes and edges, graph generative models aim to learn the distribution $p(G)$ of the observed graphs and then graph generation can be achieved by sampling a graph $G$ from the learned distribution $G \sim p(G)$.

According to the size of generated graph, graph generation tasks can be classified into two categories: (1) *fixed-size* generation in which the number of nodes is fixed across different graph samples; For example, in human brain networks (e.g., functional connectivity), the number of brain regions is

usually the same across different human subjects; and (2) *variable-sized* generation when the number of nodes varies across graph samples. For example, different molecules can be considered as graphs with various numbers of atoms. The two categories are accommodated with different types of datasets. Recent studies on graph generation could be divided into two branches, (1) one-shot generation, (2) sequential generation, based on the their choices of the generation process. Specifically, one-shot generation builds probabilistic matrices for the generated graph features which the graph structures could be obtained by taking the maximum probability nodes and edges in one shot [18, 32, 19, 33]. While sequential generation, formulates graph generation as a sequential process and generates nodes and edges one by one [34, 35, 36, 37].

## 3.2 Graph Transformation

Graph transformation aims at transforming from one graph in source domain into another graph in target domain. It can also be regarded as the graph generation conditioning on another graph. For instance, in neuroscience, it is interesting to explore the functional connectivity given the corresponding structural connectivity. In hardware design domain, given a integrated circuit design, one may be asked to obfuscate it, by adding additional gates and keys (i.e., can be considered as nodes) but maintain the same functionality. More formally, graph transformation problem can be formalized as learning a generative mapping $\mathcal{T} : (\mathcal{V}_0, \mathcal{E}_0, E_0, F_0) \rightarrow (\mathcal{V}', \mathcal{E}', E', F')$, in which $(\mathcal{V}_0, \mathcal{E}_0, E_0, F_0)$ corresponds to the graph in source domain and $(\mathcal{V}', \mathcal{E}', E', F')$ represents a graph in target domain.

Based on the entities transformed in the transformation process, problems regarding graph transformation can be divided into three main scenarios: (1) *node transformation* transforms nodes and/or their attributes from the source to the target domain; (2) *Edge transformation* maps graph topology and/or edge attributes from the source domain to the target domain; In (3) *node-edge co-transformation*, both the node and edge information can change during the transformation process.

Recent works cover each of three categories of graph transformation models. Interaction networks is a node-transformation technique that provides reasoning on objects, relations and physics [38]. DCRNN integrates diffusion convolution with a seq2seq framework to handle node transformation [39]. Graph Convolutional Policy Network is proposed for modeling chemical reactions. DCGAN has been used for generating novel protein structure [40]. GC-GAN can handle malware cyber-network synthesis [41]. For the node-edge co-transformation, JT-VAE [20] and Mol-CycleGAN [42] are designed for molecule optimization. DG-DAGRNN is employed to generalize stacked RNNs on sequences on directed acyclic graph structures [43].

# 4 Descriptions of GraphGT Benchmark Datasets

## 4.1 Taxonomy

Our GraphGT Benchmark covers 36 datasets from various domains and tasks. The taxonomy with respect to different domains is shown in Figure 2, where there are 9 domains, including protein, brain network, physical simulation, vision, molecule, transportation science, electrical and computer engineering, social network and synthetic data, across 6 subjects including biology, physics, artificial intelligence (AI), chemistry, engineering and social science. Moreover, the taxonomy by different tasks is illustrated in Figure 3. For the graph generation task, they can extract datasets for either fixed-sized generation or variable-sized generation. For the graph transformation task, we provide datasets for node transformation, edge transformation as well as node and edge co-transformation.

## 4.2 Dataset Details

In this section, we provide the specifications of representative datasets spanning different subjects introduced in Figure 2. Their potential use in tasks such as graph generation or transformation tasks will also be provided. The general profiles for different datasets are summarized in Table 1. A more detailed description of each dataset and curation method can be found in the Appendix C.

### 4.2.1 Biology

**Motivation.** In biology domain, we have two subjects which are proteins and brain networks. Proteins are essential to all lives, and are highly related to significant biomedicine-related tasks, such as protein design [57] and drug design [58, 59, 60, 61, 62, 63]. De novo protein design [64] is a promising field the explores the full sequence space which is estimated $20^{200}$ possible amino-acid sequences for only a 200-residue protein with the guidance of physical principles of protein folding. In addition to protein structure, brain networks include two major types of connectivities, structural and functional, which reflect the fiber nerve connectivity and co-activation relations, respectively, among different

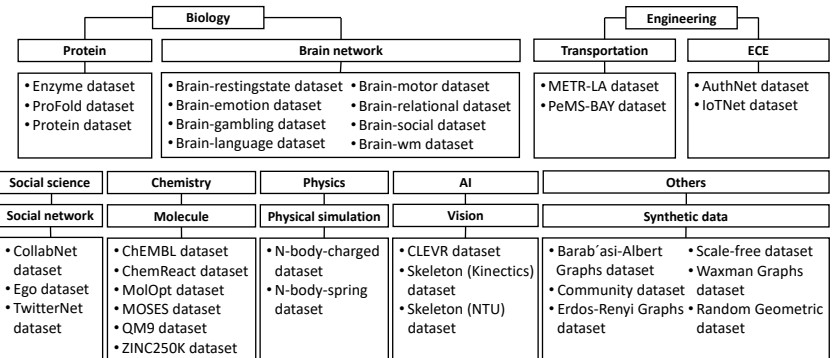

**Figure 2:** GraphGT Benchmark datasets by domains.

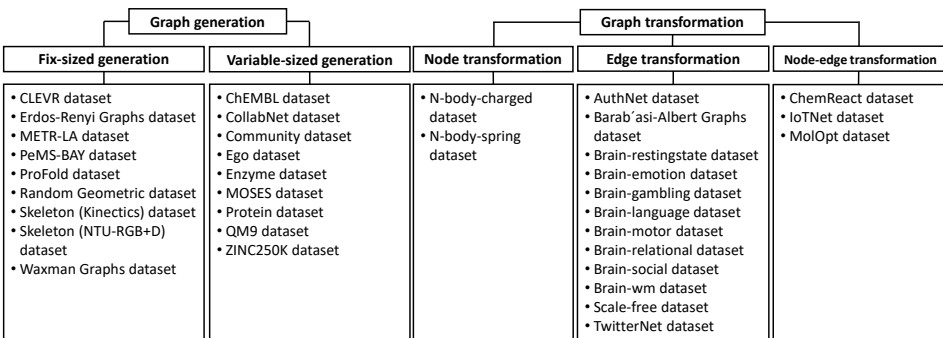

**Figure 3:** GraphGT benchmark datasets by tasks.

regions of human brains. Understanding and modeling brain networks and the correlations between structural connectivity and functional connectivity are crucial tasks in neuroscience [65].

**Tasks.** Protein structures can be considered as graphs where amino acids as nodes and contacts as edge connections. Generating novel proteins grounds up to tackle challenges in biomedicine and nanotechnology [64, 57, 58, 66, 67, 68, 67]. In a brain network, the brain regions are represented as nodes and the connectivity between each pair of regions are represented as edges. The graph transformation model can assist understanding the transformation from structural connectivity to resting-state or task-specific functional connectivities in the human brain [31].

**Dataset Construction.** We reformat 3 protein structure datasets for graph generation and 8 brain network datasets for graph transformation in GraphGT. For protein data, we start from the amino acid coordinates, and then extract graphs of protein structures according to mutual distances of amino acids. The node feature (type of amino acids) are also extracted and recorded in GraphGT. We construct 7 brain network datasets by performing standard neuroimage processing, time series processing, and network construction on both types of connectivities from the magnetic resonance imaging (MRI) data to obtain brain graphs, with edge attributes as Pearson correlation between two regions and node attributes as node index. We also reformat one brain network dataset (Brain-restingstate) that has already been used for graph transformation task [31].

### 4.2.2 Physics

**Motivation.** Physical simulation is a significant technique to explore interactions among objects with natural forces. Specific physical systems, such as dynamical systems [49], can be formed into graph structures. The dynamics of a physical system can be seen as a group of interaction components, in which complex dynamics occur at both individual level and in the system as a whole [49]. One could utilize the graph transformation methods to observe the evolution of a physical system.

**Tasks.** The dynamics of a physical system can be regarded as a graph, in which nodes represent components and edges represent their interactions. Graph transformation models have been applied to physical systems to generate possible conditions of the system sequentially [49, 69, 70]. Work in [71] utilize deep generative models to simulate physically realistic realizations of the cosmic web. Work in [72] introduces generative models in N-body simulations that pushes closer the ideas of deep generative models to practical use in cosmology.

**Table 1:** Summary of statistics and types of graphs for different GraphGT datasets. (Note: 'Y' stands for 'Yes', 'N' stands for 'No', 'GCS' stands for 'Geographic Coordinate System', '2D/3D' stands for '2D or 3D coordinates under Cartesian Coordinate System'.)

| Name | Type | #Graphs | #Nodes | #Edges | Attributed | Directed | Weighted | Signed | Homogeneous | Spatial | Temporal | Labels |
|---|---|---|---|---|---|---|---|---|---|---|---|---|
| QM9 [44] | Molecules | 133,885 | $\sim 9$ | $\sim 19$ | Y | N | Y | N | Y | 3D | N | Y |
| ZINC250K [45] | Molecules | 249,455 | $\sim 23$ | $\sim 50$ | Y | N | Y | N | Y | 3D | N | Y |
| MOSES [46] | Molecules | 193,696 | $\sim 22$ | $\sim 47$ | Y | N | Y | N | Y | 3D | N | Y |
| MolOpt [47] | Molecules | 229,473 | $\sim 24$ | $\sim 53$ | Y | N | Y | N | Y | 3D | N | Y |
| ChEMBL [48] | Molecules | 1,799,433 | $\sim 27$ | $\sim 58$ | Y | N | Y | N | Y | 3D | N | Y |
| ChemReact [31] | Molecules | 7,180 | $\sim 20$ | $\sim 16$ | Y | N | Y | N | Y | 3D | N | Y |
| Protein [30] | Proteins | 1,113 | $\sim 39$ | $\sim 73$ | Y | N | N | N | Y | N | N | Y |
| Enzyme [28] | Proteins | 600 | $\sim 33$ | $\sim 62$ | Y | N | N | N | Y | N | N | Y |
| ProFold [29] | Proteins | 76,000 | 8 | $\sim 40$ | Y | N | N | N | Y | 3D | Y | Y |
| Brain-restingstate [31] | Brain networks | 823 | 68 | 2274 | N | N | Y | Y | Y | N | N | Y |
| Brain-emotion [31] | Brain networks | 811 | 68 | 2278 | N | N | Y | Y | Y | N | N | Y |
| Brain-gambling [31] | Brain networks | 818 | 68 | 2278 | N | N | Y | Y | Y | N | N | Y |
| Brain-language [31] | Brain networks | 816 | 68 | 2278 | N | N | Y | Y | Y | N | N | Y |
| Brain-motor [31] | Brain networks | 816 | 68 | 2278 | N | N | Y | Y | Y | N | N | Y |
| Brain-relational [31] | Brain networks | 808 | 68 | 2278 | N | N | Y | Y | Y | N | N | Y |
| Brain-social [31] | Brain networks | 816 | 68 | 2278 | N | N | Y | Y | Y | N | N | Y |
| Brain-wm [31] | Brain networks | 812 | 68 | 2278 | N | N | Y | Y | Y | N | N | Y |
| N-body-charged [49] | Physical simulation networks | 3,430,000 | 25 | $\sim 3$ | Y | N | N | N | Y | 2D | Y | Y |
| N-body-spring [49] | Physical simulation networks | 3,430,000 | 5 | $\sim 10$ | Y | N | N | N | Y | 2D | Y | Y |
| CLEVR [50] | Scene graphs | 85,000 | 6 | $\sim 40$ | Y | Y | Y | N | Y | 3D | N | N |
| Skeleton (Kinetics) [51] | Skeleton graphs | 260,000 | 18 | 17 | N | N | N | N | Y | 2D | Y | Y |
| Skeleton (NTU-RGB+D) [52] | Skeleton graphs | 56,000 | 25 | 24 | N | N | N | N | Y | 3D | Y | Y |
| METR-LA [53] | Traffic networks | 34,272 | 325 | 2,369 | Y | Y | Y | N | Y | GCS | Y | Y |
| PeMS-BAY [54] | Traffic networks | 50,112 | 207 | 1,515 | Y | Y | Y | N | Y | GCS | Y | Y |
| AuthNet [41] | Authen. networks | 114/412 | 50/300 | $\sim 3/\sim 7$ | N | Y | Y | N | Y | N | N | Y |
| IoTNet [31] | IoT networks | 343 | 20/40/60 | $\sim 220/\sim 630/\sim 800$ | Y | N | Y | N | Y | N | N | Y |
| CollabNet [55] | Collab. networks | 2,361 | 303,308 | 207,632 | N | N | N | N | Y | GCS | Y | Y |
| Ego [34] | social networks | 757 | $\sim 145$ | $\sim 335$ | N | N | N | N | Y | N | N | N |
| TwitterNet [56] | social networks | 2,580 | 300 | 0.5 | N | N | N | N | Y | N | N | N |
| Barab'asi-Albert Graphs [31] | Synthetic networks | 1,000 | 20/40/60 | $\sim 60/\sim 190/\sim 300$ | Y | N | N | N | Y | N | N | N |
| Erdos-Renyi Graphs [31] | Synthetic networks | 1,000 | 20/40/60 | $\sim 100/\sim 200/\sim 400$ | Y | N | N | N | Y | N | N | N |
| Scale-Free [41] | Synthetic networks | 10,000 | 10/20/50/100/150 | 20/ 40/ 100/ 200/ 320 | N | Y | N | N | Y | N | N | N |
| Community [34] | synthetic networks | 3,000 | 64 | $\sim 340$ | N | N | N | N | Y | N | N | N |
| Random Geometric [29] | Synthetic networks | 9,600 | 25 | $\sim 350$ | Y | N | N | N | Y | Y | Y | Y |
| Waxman Graphs [29] | Synthetic networks | 9,600 | 25 | $\sim 250$ | Y | N | N | N | Y | Y | Y | Y |

**Dataset Construction.** We re-purpose two datasets that have never been tried on graph transformation tasks prior to our efforts. We start from velocities and coordinates of each particle and merge them into a single structure with node velocities as node features. Moreover, we extract temporal features from the temporal array contained in original datasets.

### 4.2.3 Artificial Intelligence

**Motivation.** Graph-structured data are widely employed in computer vision, a sub-field of AI. We store two most common graph-structured data from computer vision in GraphGT which are skeleton graphs and scene graphs. For example, generating scene graphs is of great importance to understand the relationship in a scene (i.e. image) [73]. In addition to scene graph generation, generating new human skeleton graphs also has a wide range of applications in computer vision, graphics and games, where characters could be generated and interact with human players [74, 75].

**Tasks.** In a scene graph, objects are represented as nodes and the relationship between pairs of objects is represented as edges. Graph generation models can be applied to the scene graph to help the community understand the relationship between objects in a scene, e.g. generating scene graphs with different relationships (man riding a horse vs. man standing by a horse). In a human skeleton graph, joints are represented as nodes and skeletons between each pair of joints are represented as edges. Similarly, graph generation models can be designed for skeleton graph to help the community approach interactions between human players and characters in a video (e.g. generating AI players with realistic gestures and movements).

**Dataset Construction.** We re-purpose one dataset for the scene graph and two datasets for skeleton graph that have not been used for graph generation tasks. For the scene graph, we start from the CLEVR dataset containing 10 objects in the image with different 3D locations. Then we form labeled

directed graphs with different shape of objects as the node feature and relative location between two objects as the edge feature. For skeleton graphs, we start from video clips of human action datasets, and then use OpenPose toolbox to generate skeleton with location and joints for each subject. The temporal information is also recorded and wrapped into our data as the temporal feature.

### 4.2.4  Chemistry

**Motivation.** Chemistry is another subject in which graph generation and transformation play critical roles for generating optimal molecules or predicting products of chemical reactions [20, 31, 76, 77]. The chemical space, drug-like molecules are vast and estimated to $10^{60}$ [78]. Generating novel molecules with desired properties has great potentials in discovering new drugs and materials. Modeling chemical reactions is another fundamental problem in chemistry which can advance our understanding of the properties of molecules [76].

**Tasks.** In a molecular graph, atoms are represented as nodes, and bonds are represented as edges. Molecular graph generation has numerous applications in drug discovery and [79] material science [80] to generate optimal molecules. Moreover, learning the transformation from the reactants to the products can help the community better understand the mechanism of chemical reactions [76].

**Dataset Construction.** We reformat 6 datasets in chemistry by converting SMILES sequence into molecular structures. Then the molecular structures are converted into graphs with atoms as nodes and chemical bonds as edges. Atom and bond type serve as node and edge feature respectively.

### 4.2.5  Engineering

**Motivation** For the engineering field, we provide datasets corresponding to two domains, transportation system and electrical and computer engineering (ECE). First of all, a few graph representation learning methods such as graph neural networks have been applied to transportation research such as traffic prediction [81, 39]. In addition to graph representation learning tasks, graph generative models in machine learning have started experiencing increase in recent years, for tasks like human mobility generative modeling [82] given that a number of tasks can be formalized into a graph generation or transformation problem in the field of engineering. The road system can also be considered as graphs where road segments and interactions are connected, for which the graph generative models can be employed for generating newly designed networks [83].

**Tasks.** In internet network, graphs contain nodes representing devices, and edges representing connection between two devices. The malware confinement over the internet can be treated as a graph transformation problem to generate optimal status of network that limits malware propagation [31]. Traffic networks contain graphs with nodes as speed sensors and edges as roads. Traffic networks can be employed with graph generation models for designing new and efficient traffic networks.

**Dataset Construction.** We reformat the malware dataset by adopting the initial attacked networks (i.e., the Internet of Things) as the input graphs, with nodes representing devices and edges representing their connections. Malware confinement status are extracted as node features and distances between two devices are edge features. We also split the dataset according to their graph sizes for different graph transformation purposes. We reformat two transportation datasets by extracting them from LA-Metro and PeMS projects, respectively. We extract sensors as graph nodes an roads as edges, with traffic speed as the node feature. We also extract GCS spatial features and temporal features in the dataset.

### 4.2.6  Social Science

**Motivation.** Social networks are an important type of graphs where people or other subjects are connected by relationships such as friendship and co-authorship, and have been widely explored in social science, statistics, and physics with network (generative) modeling techniques. The advancement of graph generative models further stimulate the social network research by handling different aspects of the data. For example, DYMOND achieves graph generation on social networks by borrowing building blocks of network structure to capture long-range interactions [84]. Another graph generative model, TagGen, can preserve both structural and temporal information in the process of modeling interactions in the social network [85].

**Tasks.** Social networks can be formalized into graphs in which social subjects are nodes and their relationships are edges. The community network has been used to on graph generative models so that the relationship between people or community could be modeled and understood [34].

**Dataset Construction.** We reformat Ego dataset from Citeseer dataset. Nodes represent documents and edges represent citation relationships. We also re-purpose TwitterNet from [56]. Both datasets do not have node or edge attributes. We construct from scratch the graphs of CollabNet by selecting

authors as nodes and co-authorships as edges. To cut the graphs into pieces, we generate sub-graphs based on the fields of study of papers. For each field, we generate one spatio-temporal graphs.

### 4.2.7 Synthetic

**Motivation.** The limited amount of available data in the real world, especially graph data for specific geometric properties [86, 87, 88] for graph generation and transformation problems, limits the advance of the field. Synthetic data is a way to overcome this obstacle and prolong the march of progress in graph generation and transformation tasks. This motivate us to reformat a few simulated synthetic datasets in GraphGT to accommodate various needs of the community for evaluating graph generation and transformation tasks.

**Tasks.** Synthetic datasets contain graphs corresponding to various geometric properties, including scale-free graphs, Erdos-Renyi graphs, random geometric graphs and so on. A huge amount of works regarding graph generation and transformation have been using synthetic datasets to evaluate their models. NEC-DGT is evaluated with Barab'asi-Albert graphs and Erdos-Renyi graphs [31]. Another graph transformation model, GT-GAN, is evaluated by scale-free graphs [41].

**Dataset Construction.** We reformat synthetic datasets by converting the original sparse matrices into dense matrices, and reshaping them into predefined dimensions. All synthetic datasets are simulated based on specific geometric properties or laws.

## 5 Benchmark Experiments

### 5.1 Graph Generation

#### 5.1.1 Evaluation Metrics

The evaluation of graph generation performance has been widely recognized as a challenging tasks [34, 37] and there lacks a unified framework that can provide comprehensive evaluation procedures and metrics. Following the survey of graph generation [7], we enhanced our deployed API with easy-to-use evaluation tools. The evaluation metrics in GraphGT is elaborated as follows.

In **statistics-based** evaluation metrics, the quality of the generated graphs is accessed by computing the distance between the graph statistic distribution of real graphs and generated graphs. In the deployed API, seven typical graph statistics are considered, which are summarized as follows: (1) *Node degree distribution*: the empirical node degree distribution of a graph, which could encode its local connectivity patterns. (2) *Clustering coefficient distribution*: the empirical clustering coefficient distribution of a graph. Intuitively, the clustering coefficient of a node is calculated as the ratio of the potential number of triangles the node could be part of to the actual number of triangles the node is part of. (3) *Orbit count distribution*; the distribution of the counts of node 4-orbits of a graph. Intuitively, an orbit count specifies how many of these 4-orbits substructures the node is part of. This measure is useful in understanding if the model is capable of matching higher-order graph statistics, as opposed to node degree and clustering coefficient, which represent measures of local (or close to local) proximity. (4) *Largest connected component*: the size of the largest connected component of the graphs. (5) *Triangle count*: the number of triangles counted in the graph. (6) *Characteristic path length*: the average number of steps along the shortest paths for all node pairs in the graph. (7) *Assortativity*: the Pearson correlation of degrees of connected nodes in the graph. To calculate the distances regarding the above mentioned statistics, *Average Kullback-Leibler Divergence* and *Maximum Mean Discrepancy (MMD)* are utilized.

In **self-quality based** evaluation, the quality of the generated graphs, validity, uniqueness and novelty, are measured. The definition and calculation of the three metrics are provided as follows: (1) *Validity*: validity evaluates graphs by judging whether they preserve specific properties. For example, for cycles graphs/tree graphs, the validity is calculated as the percentage of generated graphs that are cycles or trees [8]. For molecule graphs, validity is the percentage of chemically valid molecules based on domain-specific rules [36]. (2) *Uniqueness*: ideally, high-quality generated graphs should be diverse and similar, but not identical. Thus, uniqueness is utilized to capture the diversity of generated graphs [89, 8, 36]. To calculate the uniqueness of a generated graph, the generated graphs that are sub-graph isomorphic to some other generated graphs are first removed. The percentage of graphs remaining after this operation is defined as Uniqueness. For example, if the model generates 100 graphs, all of which are identical, the uniqueness is $1/100 = 1\%$. (3) *Novelty*. Novelty measures the percentage of generated graphs that are not sub-graphs of the training graphs and vice versa [89]. Note that identical graphs are defined as graphs that are sub-graph isomorphic to each other. In other words, novelty checks if the model has learned to generalize unseen graphs.

**Table 2:** Quantitative evaluation and comparison on spatial network generation tasks by different deep generative models on graphs ("Deg." is short for degree distribution. "Clus." is short for clustering coefficient distribution. "Orbit." is short for average orbit counts statistics. ).

| Method → | GraphRNN | | | GraphVAE | | | GraphGMG | | |
|---|---|---|---|---|---|---|---|---|---|
| Dataset ↓ | Deg. (%) | Clus. (%) | Orbit. (%) | Deg. (%) | Clus. (%) | Orbit. (%) | Deg. (%) | Clus. (%) | Orbit. (%) |
| Waxman | 1.20 | 1.74 | 0.87 | 120.14 | 144.22 | 109.72 | 26.44 | 41.58 | 21.15 |
| Random Geometric | 1.09 | 19.19 | 2.80 | 88.27 | 95.52 | 102.71 | 57.12 | 111.94 | 71.32 |
| CLEVR | 56.89 | 2.66 | 61.19 | 0.00 | 0.00 | 0.00 | 126.96 | 163.53 | 180.65 |
| METR-LA | 193.11 | 196.69 | 165.86 | - | - | - | - | - | - |
| PeMS-BAY | 172.97 | 173.37 | 159.68 | - | - | - | - | - | - |
| ProFold | 1.10 | 0.38 | 0.09 | 114.60 | 109.02 | 84.78 | 5.55 | 44.61 | 4.55 |
| Skeleton (Kinetics) | $< 10^{-5}$ | 0.00 | $< 10^{-5}$ | 200.00 | 200.00 | 200.00 | 9.84 | 0.00 | 0.06 |
| Skeleton (NTU-RGB+D) | $< 10^{-5}$ | 0.00 | $< 10^{-5}$ | 200.00 | 200.00 | 200.00 | 120.31 | 0.27 | 2.31 |
| CollabNet | - | - | - | - | - | - | - | - | - |
| N-body-charged | 172.93 | 0.00 | 0.00 | 0.00 | 0.00 | 0.00 | 37.83 | 75.48 | 2.76 |
| N-body-spring | 3.17 | 1.86 | 0.02 | 141.06 | 123.22 | 5.71 | 127.42 | 49.46 | 0.75 |
| Ego | 66.44 | 129.82 | 64.18 | - | - | - | - | - | - |
| Community | 19.61 | 55.46 | 57.09 | - | - | - | - | - | - |
| Protein | 2.57 | 5.27 | 1.27 | - | - | - | - | - | - |
| Enzyme | 0.81 | 1.64 | 0.88 | - | - | - | - | - | - |

### 5.1.2 Benchmark Results

For graph generation, we benchmark 16 graph generation datasets in GraphGT with GraphRNN [34], GraphVAE [18], and GraphGMG [8], three common graph generation baselines. The detailed descriptions of each baseline models can be found in Appendix D. We evaluate the performance of the graph generative models on three statistics-based metrics, degree distribution, clustering coefficient distribution and orbit counts statistics. For efficiency problem, GraphVAE and GraphGMG cannot scale to multiple large datasets, e.g. METR-LA, Protein, Enzyme, etc. Note that the CollabNet is too large even for GraphRNN to scale. From Table 2, we can observe that GraphRNN outperforms GraphVAE and GraphGMG in most of the datasets. Notably, GraphRNN takes the advantage of sequential graph generation which allows scaling to large graphs, while GraphVAE cannot due to its costly one-shot generation method. Additioanlly, GraphRNN works extraordinarily well on relatively small graphs datasets, e.g. Profold, N-body, Skeleton, while performs worse on large graphs like traffic networks. GraphVAE performs very well in two particular datasets which are CLEVR and N-body-charged which both of them are very small and the simulation processes are stochastic. GraphGMG performs well in specifically one skeleton graph and one protein dataset which both of the graph structures are fixed and simple. Additionally, GraphVAE outperforms the sequence-based models on CLEVR and N-body-charged datasets. We believe that it is easier for an one-shot generation method to learn topology which is related to spatial locations since it doesn't have to learn a sequence-dependent process.

## 5.2 Graph Transformation

### 5.2.1 Evaluation Metrics

In **Graph-property-based** evaluation, we directly compare each generated graph to its target graph via the following metrics: (1) random-walk kernel similarity by using the random-walk based graph kernel [90]; (2) combination of Hamming and Ipsen-Mikhailov distances(HIM) [91]; (3) spectral entropies of the density matrices; (4) eigenvector centrality distance [92]; (5) closeness centrality distance [93]; (6) Weisfeiler Lehman kernel similarity [94]; (7) Neighborhood Sub-graph Pairwise Distance Kernel [95] by matching pairs of subgraphs with different radii and distances; (8) Jensen–Shannon distance, (9) Bhattacharyya distance and (10) Wasserstein distance by measuring distance of node degrees of two graphs.

In **Mapping-relationship-based** evaluation, we measure whether the learned relationship between the input and the generated graphs is consistent with the true relationship between the input and the real graphs. There are two kinds of relationship to be considered [7]: (1) *Explicit mapping relationship*. Considering the situation where the true relationship between the input conditions and the generated graphs is known in advance, the evaluation can be conducted as follows: we quantitatively compare the property scores of the generated and input graphs to see if the change indeed meets the requirement. For example, one can compute the improvement of logP scores from the input molecule to the optimized molecule in molecule optimization task [96]. (2) *Implicit mapping relationship*. When the underlying patterns of the mapping from the input graphs to the real target graphs are implicit and complex to define and measure, a classifier-based evaluation metric can be utilized [41]. By regarding the input and target graphs as two classes, it assumes that a classifier that is capable of distinguishing the generated target graphs would also succeed in distinguishing the real target graphs from the input graphs. Specifically, a graph classifier is first trained based on the input

**Table 3:** Quantitative evaluation and comparison on transformation tasks by different deep transformation models on graphs ("JS-dist." is the Jensen–Shannon distance. "BH-dist." is the Bhattacharyya distance. "WS-dist." is the Wasserstein distance.).

| Method → | Interaction Network | | | NEC-DGT | | |
|---|---|---|---|---|---|---|
| Dataset ↓ | JS-dist. (%) | BH-dist. (%) | WS-dist. (%) | JS-dist. (%) | BH-dist. (%) | WS-dist. (%) |
| AuthNet | 1.04 | 0.01 | 0.33 | 82.81 | 95.88 | 24.59 |
| Barab'asi-Albert Graphs | 4.50 | 0.21 | 5.12 | 66.87 | 59.39 | 36.84 |
| Brain-restingstate | 11.17 | 1.26 | 13.26 | 11.39 | 1.31 | 18.24 |
| Brain-emotion | 12.63 | 1.61 | 15.78 | 12.83 | 1.66 | 12.58 |
| Brain-grambling | 12.55 | 1.59 | 15.73 | 12.82 | 1.66 | 26.54 |
| Brain-language | 12.23 | 1.51 | 15.24 | 12.56 | 1.60 | 16.51 |
| Brain-motor | 11.88 | 1.43 | 14.69 | 12.14 | 1.49 | 31.04 |
| Brain-relational | 12.26 | 1.52 | 15.23 | 12.50 | 1.58 | 35.62 |
| Brain-social | 12.09 | 1.48 | 14.97 | 12.34 | 1.54 | 141.58 |
| Brain-wm | 12.23 | 1.51 | 15.24 | 12.48 | 1.58 | 37.31 |
| Scale-free | 1.19 | 0.01 | 0.42 | 79.13 | 83.00 | 21.71 |
| TwitterNet | 0.01 | $< 10^{-3}$ | $< 10^{-3}$ | $< 10^{-3}$ | $< 10^{-3}$ | 6155.10 |
| N-body-charged | 0.12 | $< 10^{-3}$ | 0.14 | 4.37 | 0.21 | 47.52 |
| N-body-spring | 0.05 | $< 10^{-3}$ | 0.07 | 4.50 | 0.20 | 53.20 |
| ChemReact | 0.94 | $< 10^{-3}$ | 0.27 | 77.84 | 79.92 | 0.6714 |
| IoTNet | 17.01 | 3.01 | 19.32 | 65.39 | 55.90 | 2572.62 |
| MolOpt | 0.71 | 0.01 | 0.11 | 82.67 | 94.89 | 19.97 |

and generated target graphs. Then this trained graph classifier is tested to classify the input graph and real target graphs, and the results will be used as the evaluation metrics.

### 5.2.2 Benchmark Results.

Here, 17 transformation datasets are benchmarked for graph transformation tasks in GraphGT. Two state-of-the-art graph transformation models, Interaction network (IN) [38] and Node-Edge Co-evolving Deep Graph Translator (NEC-DGT) [31] are borrowed to analyze these datasets. Three metrics, Jensen–Shannon distance, Bhattacharyya distance and Wasserstein distance, are used to measure the distance between the distribution of generated graphs and target graphs. Details regarding the experimental settings can be found in Appendix D. We find that two models have a close performance regarding graph transformation on most datasets. This is not surprising since two models follow similar philosophies to handle node interactions in the graph. With the Interaction Network, the smallest Jensen–Shannon and Bhattacharyya distance are achieved on TwitterNet, which is aligned with NEC-DGT. TwitterNet also has the closest Wasserstein distance, whether Brain-emotion has the closest Wasserstein distance for NEC-DGT. This difference might originate from the capacity to handle node or edge features of two models, or different hyper-parameter settings. Interaction Networks can handle edge attributes, which are available for Brain-emotion dataset but not for TwitterNet dataset, whereas NEC-DGT can handle both node and edge attributes, neither of which are available for TwitterNet. We also find that, for the same model, datasets from different domains have different performances. We observe a relatively large distances regarding three metrics for 8 brain network datasets compared with most other datasets when being evaluated by Interaction Network. However, these 8 datasets have a relatively smaller distance when being evaluated by NEC-DGT. This reflects the complexity of the brain network domain [97] that needs more advanced models to be handled, such as NEC-DGT. N-body-charged and N-body-spring datasets have a generally smaller distances compared with most other datasets when being evaluated by both models. This results from the relatively small graph size in physical simulation domain (Table 3).

## 6 Conclusion

We introduce GraphGT, a large dataset collection for graph generation and transformation problems. GraphGT covers datasets in 9 domains across 6 subjects, in which CollabNet dataset and 7 brain network datasets are collected and constructed from scratch for graph generation and transformation. Another 8 datasets are re-purposed by us from other applications into graph generation and transformation tasks for the first time. The remaining are from very different domains that share quite different terminology, formats, and data structures, which are reformatted by us to a unified format for the first time for easy access and use in a standardized manner. In addition, we provide 3 types of Python APIs, including dataset downloader, graph generation data processor, graph transformation data processor and evaluator, for users to query and access datasets according to specific disciplines, domains and applications per their interests. Finally, we provide 16 graph generation benchmark results and 17 graph transformation benchmark results We believe that GraphGT can advance the community to address significant challenges in graph generation and transformation.

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
