# A  Key Information about GraphGT

## A.1  Dataset Documentation

We provide detailed documentation of dataset collection, processing, task for each dataset both in section C and in our website. We provide statistics, taxonomy, detailed description, and task for each dataset and can be tracked in our website https://graphgt.github.io/.

## A.2  Intended Use

GraphGT is intended for the deep graph learning as well as specific domain (e.g. physics, biology, chemistry, etc.) community to use and develop machine learning algorithms to advance applications in various domains.

## A.3  URLs

Official website (https://graphgt.github.io/) contains all references of GraphGT, including dataset taxonomy, task, evaluation, visualization, tutorials, papers, GitHub, and other useful resources. GitHub repository (https://github.com/yuanqidu/GraphGT) hosts all source codes, installation instructions, and tutorials of GraphGT.

## A.4  Hosting and Maintenance Plan

Our GraphGT Python library is regularly maintained and version-tracked via GitHub. All datasets are currently hosted on Dropbox and will be transferred to Emory University server soon. Our dataset is both directly downloadable with a Dropbox link or from our Python APIs. Our core team commit to maintain this initiative for at least five years. In the meantime, we will expand the community in multiple dimensions and attract external contributors from the whole community. We will regularly update new dataset, task, evaluation and visualization methods to GraphGT.

## A.5  Limitations

Graph generation and transformation is a fast-growing, vast, and promising field and their applications cover a wide range of applications. We start this initiative to build the infrastructure for the community which includes most of the mainstream datasets in the graph generation and transformation field and many more new datasets. However, it is an ongoing effort and we strive to continuously include more datasets, evaluation and visualization methods to advance the field.

## A.6  Potential Negative Societal Impacts

Graph generation and transformation are motivated by generating novel graph-structured data and understanding the graph-structured data; thus, they have vast applications, such as drug discovery, protein design, mobility synthesis, etc., which could potentially lead to better designed drug, traffic network, etc., and save lives, time, etc. We envision that GraphGT can facilitate algorithmic and scientific advances in various domains across subjects and accelerate machine learning model development and application for real-world use. GraphGT neither involves human subject research nor contains personally identifiable information.

# B  Dataset Format

We store each of the dataset in a Numpy[3] array format. For different datasets with different information available as shown in Table 1. For all the datasets, each has at most five types of features available including adjacency matrices, node features, edge features, spatial features, and labels. Among all the features, *adjacency matrices* denote the edge connections between pairs of nodes, *node features* denote features attaching to each node, *edge features* denote features attaching to each edge connection, *spatial features* denote the spatial geometry of a graph (in most of the cases, they are coordinates attaching to each node), *labels* denote either node-level or graph-level labels of a graph. For temporal graphs, we store two versions of the graphs, which one flattens and shuffles all the snapshots of the temporal graphs, and the other one keeps the temporal dimension and order. For graph transformation datasets, we store both the source and the target graph and available features separately.

# C  Dataset Details

We list detailed information for each of the datasets in GraphGT.

---

[3]https://numpy.org/doc/

## C.1 Molecules

We have 6 molecule datasets, in which 4 (QM9 [44], ZINC250K [45], MOSES [46], ChEMBL [48]) for graph generation and 2 (MolOpt [47], ChemReact [31]) for graph transformation. For all of the molecule datasets, we store adjacency matrix, node feature (i.e. atoms), edge feature (i.e. bonds), spatial feature (i.e. geometry), and smiles (i.e. string representation). There are in total 4 types of atoms in QM9, 0 = H, 1 = C, 2 = N, 3 = O, 4 = F. There are in total 14 types of atoms in ZINC250K dataset, MOSES, and ChEMBL dataset, 0 = Br, 1 = C, 2 = Cl, 3 = F, 4 = H, 5 = I, 6 = N, 7 = N, 8 = N, 9 = O, 10 = O, 11 = S, 12 = S, 13 = S. There are in total 4 types of bonds in all the datasets, and we represent them as follows: 0 = Single, 1 = Double, 2 = Triple, 3 = Aromatic.

**QM9** [44] dataset is an enumeration of around 134,000 stable organic molecules with up to 9 heavy atoms (carbon, oxygen, nitrogen and fluorine). As no filtering is applied, the molecules in this dataset only reflect basic structural constraints. In QM9 dataset, each graph contains approximately 9 nodes and 19 edges. A node in QM9 represents an atom with atom type as the node feature. An edge in QM9 dataset represents a bond in the molecule with bond type as the edge feature. Moreover, QM9 dataset contains the 3D spatial feature for each graph. In GraphGT, the QM9 dataset has been reformatted as adj.npy, edge_feat.npy, label.npy, node_feat.npy and spatial.npy that contain molecular structure information, node features, edge features and spatial features.

The information of QM9 is initially stored in .xyz files separately for each molecule. We use Python to process the SMILE of each molecule and convert the molecule graph to Numpy formats.

**ZINC250K** [45] dataset is a curated set of 250k commercially available drug-like chemical compounds. On average, these molecules are bigger (about 23 heavy atoms) and structurally more complex than the molecules in QM9 dataset. Each graph in ZINC250K dataset contains approximated 23 nodes and 50 edges. In ZINC250K dataset, each node represents an atom, with atom type as the node feature. An edge in 250K dataset represents a bond in the molecule with bond type as the edge feature. 250K dataset also contains 3D spatial feature for each graph. In GraphGT, the ZINC250K dataset has been reformatted as adj.npy, edge_feat.npy, label.npy, node_feat.npy and spatial.npy that contain molecular structure information, node features, edge features and spatial features.

ZINC dataset is stored in one .csv file including 249,455 molecules. After reading the data by Python, we process the SMILE of each molecule to convert the data to a graph. And all the graphs are saved in .npy format.

**Molecular Sets (MOSES)** [46] is a benchmark platform for distribution learning based molecule generation. Within this benchmark, MOSES provides a cleaned dataset of molecules that are ideal of optimization. It is processed from the ZINC Clean Leads dataset, and contains 193,696 molecules in total. Each graph in the dataset contains around 22 nodes and 47 edges. In MOSES dataset, each node represents an atom, with atom type as the node feature. An edge in MOSES dataset represents the bond in the molecule with bond type as the edge feature. MOSES datasets also contains 3D spatial features.

The data is originally stored in a .txt file. We first read the data and then process the SMILE of the molecule based on the Python rdkit library. The final data format is saved as .npy files.

**ChEMBL** [48] dataset is a manually curated database of bioactive molecules with drug-like properties. It brings together chemical, bioactivity and genomic data to aid the translation of genomic information into effective new drugs. ChEMBL contains 1,799,433 graphs in total. Each graph in the dataset contains around 27 nodes and 58 edges. In ChEMBL dataset, each node represents an atom, with atom type as the node feature. An edge in ChEMBL dataset represents the bond in the molecule with bond type as the edge feature. This datasets also contains 3D spatial features.

ChEMBL is originally stored in a .txt file containing all the molecules. We first read the data and then process the SMILE of the molecule based on the Python rdkit library. The final data format is saved as .npy files.

**MolOpt** [47] dataset extracts translation pairs from the ZINC database in terms of three molecular properties, Penalized logP, Drug-likeness, and Dopamine Receptor. MolOpt contains 229,473 pairs of graphs in total. Each graph in the dataset contains around 24 nodes and 53 edges. In MolOpt dataset, each node represents an atom, with atom type as the node feature. An edge in ChEMBL dataset represents the bond in the molecule with bond type as the edge feature. This datasets also contains 3D spatial features.

This dataset is originally stored in several .csv files and the format of the dataset has been preprocessed. We read the .csv files and convert the SMILE molecules to graphs and then save them as .npy files.

**ChemReact** [31] dataset has totally 7180 pairs of reactant and product molecule graph in the dataset derived from USPTO dataset. Each graph in the dataset contains around 20 nodes and 16 edges. In ChemReact dataset, each node represents an atom, with atom type as the node feature. An edge in ChemReact dataset represents the bond in the molecule with bond type as the edge feature. This datasets also contains 3D spatial features. [95].

Chemical Reaction dataset is originally stored in several .txt files. The first step for processing the data is to aggregate data from different sources. Then we convert the SMILE of molecules to graph formats, and then save them in .npy files.

### C.1.1   License

**QM9**: CC BY-NC-SA 4.0.

**ZINC250K**: Free to use for everyone.

**MOSES**: The dataset is generated by [46], which is under MIT License. The license of the dataset is not specified.

**ChEMBL**: CC BY-NC-SA 3.0.

**MolOpt**: Extracted from ZINC Database.

**ChemReact**: Not specified.

### C.2   Proteins

We have three protein datasets available in GraphGT, which includes protein structures, Enzyme and dynamic protein folding process.

**Protein** [30] dataset contains 918 protein graphs. Each protein is represented by a graph in Protein dataset, where nodes are amino acids and two nodes are connected if they are less than 6 Angstroms apart. Proteins dataset contains 1,113 graphs in total. Each graph in the dataset contains around 39 nodes and 73 edges. Node feature is contained in the dataset representing the type of amino acids. Protein dataset can be used for attributed graph generation.

Protein dataset is originally stored in several .txt files with the unit of node. We read all .txt files to generate graphs, convert them to Numpy arrays and save them in .npy format.

**Enzyme** [28] dataset contains protein tertiary structures representing 600 Enzyme. Nodes in a graph (protein) represent secondary structure elements, and two nodes are connected if the corresponding elements are interacting. The node labels indicate the type of secondary structure, which is either helices, turns, or sheets. Each graph in the dataset contains around 33 nodes and 62 edges. The node features in the graph represent type of amino acids. This dataset can be employed for attributed graph generation.

Enzyme dataset is originally stored in several .txt files with the unit of node. We read all .txt files to generate graphs, convert them to Numpy arrays and save them in .npy format.

**ProFold** [29] dataset contains dynamic folding processes of a protein peptide with sequence AGAAAAGA in 38 steps. ProFold contains 76,000 graphs in total. Each graph has 8 nodes and around 40 edges. The node represents amino acid of the protein, and the edge represent the bond between amino acids. The node feature of each protein is the sequence (AGAAAAGA) along with the spatial locations of each amino acid, and the edge feature of each protein is an adjacency matrix constructed by connecting all pairs of nodes with distance $< 8$ Å. This dataset can be used for either attributed graph generation or temporal graph generation.

### C.2.1   License

**Enzyme**: CC-BY-4.0.

**ProFold**: The dataset is collected by [29]. The license is not specified.

**Protein**: CC-BY-4.0.

### C.3   Brain Networks

The Brain dataset comes from the human connectome project (HCP) [31] and has a few branches: restingstate, emotion, gambling, language, motor, relational, social and wm according to different tasks. In this dataset, the source graphs reflect the structural connectivity (SC), and the target graphs represent the functional connectivity [31]. Specifically, both types of connectivities are processed from the magnetic resonance imaging (MRI) data from HCP. SC is obtained by applying probabilistic tracking on the diffusion MRI data by Probtrackx tool from the FMRIB Software Library [96] with 68 regions of insterest (ROI). The edge attributes of FC are defined as Pearson's correlation between two ROIs blood oxygen level-dependent time obtained from the resting-state functional MRI data.

Node attributes is a one-hot vector representing index of each node. In total, 823 pairs of SC and FC samples are enrolled in the dataset. The dataset has been splitted into 8 categories for 8 specific domains, including Brain-restingstate, Brain-emotion, Brain-gambling, Brain-language, Brain-motor, Brain-relational, Brain-social and Brain-wm. All of these datasets can be employed for eight weighted graph transformation or signed graph transformation tasks.

Originally, data is a group of .npz files, containing the structural connectivities for each subject, functional connectivities for each subject, and list of subject IDs for each task using different correlations. Unfortunately, the subjects used are not universal for all tasks, and so we eliminate all but those that appeared in every single task. From there, we simply concatenate all of the functional connectivities from all of the various tasks using FC correlation, and concatenated all of the structural connectivities from all of the various tasks using FC correlation, thus creating FC_concatenated_edge_feat and SC_concatenated_edge_feat. For the adjacency matrix containing .npy arrays, we encounter a small issue; the adjacency matrix is required to be formatted with a specific shape, but that shape is not compatible with the edge feature shape, and so we make the adjacency matrix a placeholder basically. For details please refer to readme.txt.

### C.3.1   License

**Brain**: This dataset comes from the human connectome project. Data collection and sharing for this project was provided by the MGH-USC Human Connectome Project (HCP; Principal Investigators: Bruce Rosen, M.D., Ph.D., Arthur W. Toga, Ph.D., Van J. Weeden, MD). HCP funding was provided by the National Institute of Dental and Craniofacial Research (NIDCR), the National Institute of Mental Health (NIMH), and the National Institute of Neurological Disorders and Stroke (NINDS). HCP data are disseminated by the Laboratory of Neuro Imaging at the University of Southern California.

### C.4   Physical Simulations

**N-body-charged** [49] dataset simulates a system containing 5 particles with positive or negative charges. Particles are located in 2D coordinates without any external forces except attracting force and repelling force. The quantity of electrical charges is sampled from uniform probability. Each particle interacts via Coulomb forces. Every two particles interact, either attract or repel each other. The temporal length of each sequence is 49, which obtains from sub-sampling every 100 steps in a trajectory. N-body-charged dataset contains 3,430,000 graphs in total, each of which contains 25 nodes with around 3 edges. Each node represents a particle and each edge represents interaction between nodes. Node attribute represents node input. 2d spatial features and temporal are included in the dataset. N-body-charged can be used for either attributed graph transformation, spatial graph transformation or temporal graph transformation.

Originally, for the charged dataset, there are separate numpy files for the velocities, edges, and locations of each particle for train, validation, and testing. Then, all velocity arrays(train, valid, test) for the charged dataset were merged into a single one, and the same was done for all of the location arrays, and all of the edge arrays. To convert the charged edge features into adjacency matrices, all nonzero values were turned to ones, and since all particles had some form of connection, that meant all adjacency matrices ended up being all ones for the charged dataset. Then, for each new temporal array we had here, we created a new version: a non-temporal one, where we concatenated the first two dimensions of the array, as the second dimension represented the different temporal instances. For details information, please refer to readme.txt.

**N-body-spring** [49] dataset simulates a system containing 5 particles connected by springs. Particles are located in 2D coordinates without any external forces except elastic collisions. Particles are connected via springs with probability of 0.5, and interactions between springs follow Hooke's law. The initial location of each particle is sampled from a Gaussian distribution and the initial velocity of each particle is a random vector of norm 0.5. The trajectories of all springs are calculated by solving Newton's equations of motion PDE. The temporal length of each sequence is 49, which obtains from sub-sampling every 100 steps in a trajectory. N-body-spring dataset contains 3,430,000 graphs in total, each of which contains 5 nodes with around 10 edges. Each node represents a particle and each edge represents interaction between nodes. Node attribute represents node input. 2D spatial features and temporal features are included in the dataset. N-body-spring can be used for either attributed graph transformation, spatial graph transformation or temporal graph transformation.

Originally, for the spring dataset, there were separate numpy files for the velocities, edges, and locations of each particle for train, validation, and testing. There are 5 particles, 5 springs in each graph. Then, all velocity arrays(train, valid, test) for the spring dataset were merged into a single one,

and the same was done for all of the location arrays, and all of the edge arrays. For the springs dataset, we had only ones and zeroes in the edges: connection or no connection, and so we simply took this as our adjacency matrix as well for each matrix in the springs dataset. Then, for each new temporal array we had here, we created a new version: a non-temporal one, where we concatenated the first two dimensions of the array, as the second dimension represented the different temporal instances.

### C.4.1 License

**N-body-charged**: The dataset is simulated by [49], which is under MIT License. The license of the dataset is not specified.

**N-body-spring**: The dataset is simulated by [49], which is under MIT License. The license of the dataset is not specified.

### C.5 Collaboration Networks

**CollabNet** [55] dataset is collected from DBLP-Citation-network V12, which contains around 4.9 million papers and 45 million citation relationships. We construct graphs by selecting authors as nodes and co-authorships as edges during the time period from 1990 to 2019. To cut the graphs into pieces, we generate sub-graphs based on the Fields of Study attribute from papers. For each field, we generate one spatio-temporal graph. We generate 2361 spatio-tempora graphs with a total of 303,308 nodes and a total of 207,632 of edges. This dataset contains temporal and GCS spatial features, so that the dataset can be used for spatial graph generation and temporal graph generation.

### C.5.1 License

**CollabNet**: The dataset is collected from DBLP-Citation-network V12. The license is not specified.

### C.6 Traffic Networks

**METR-LA** [53] dataset is collected by Los Angeles Metropolitan Transportation Authority (LA-Metro), and processed by University of Southern California's Integrated Media Systems Center. This dataset contains traffic information collected from 207 loop detectors in the highway of Los Angeles County for 4 months (from Mar 1st 2012 to Jun 30th 2012). Each sensor records traffic speed value per 5 minutes. The dataset contains 34,272 graphs, each of which has 325 nodes and 2,369 edges. In METR-LA, each node represent a speed senor and each edge represents a road. The node features of the dataset represent the traffic speed captured by the sensor. The dataset contains GCS spatial features and temporal features. METR-LA can be used for spatial graph generation, temporal graph generation, attributed graph generation and weighted graph generation.

The information of the METR-LA dataset is stored in three files with different formats. We borrow Python to read these data, and convert them to Numpy formats. We then save the data in .npy format.

**PeMS-BAY** [54] dataset is collected by California Transportation Agencies (CalTrans) Performance Measurement System (PeMS). PeMS-BAY dataset collects traffic information in the Bay Area. The dataset contains traffic information of 325 sensors within 5 months (From Jan 1st 2017 to May 31st 2017). Each sensor records traffic speed value per 5 minutes. The dataset contains 50,221 graphs, each of which has 207 nodes and 1,515 edges. In PeMS-BAY, each node represent a speed senor and each edge represents a road. The node features of the dataset represent the traffic speed captured by the sensor. The dataset contains GCS spatial features and temporal features. PeMS-BAY can be used for spatial graph generation, temporal graph generation, attributed graph generation and weighted graph generation.

The information of the PeMS-BAY dataset is stored in three files with different formats. We borrow Python to read these data, and convert them to Numpy formats. We then save the data in .npy format.

### C.6.1 License

**METR-LA**: The dataset is collected by Los Angeles Metropolitan Transportation Authority (LA-Metro), and processed by University of Southern California's Integrated Media Systems Center. The license is not specified.

**PeMS-BAY**: The dataset is collected by California Transportation Agencies (CalTrans) Performance Measurement System (PeMS). The license is not specified.

### C.7 Authentication Networks

**AuthNet** dataset includes the authentication activities of users on their computers and servers in their enterprise computer network and is published by Los Alamos National Laboratory (LANL). [97, 41]. There are two subsets of different sizes of graphs (e.g., 50 and 300) in AuthNet dataset with 114 and 412 graphs, respectively. For each subset, we train and test folder separately. Train set contains the

graph pairs (one-to-one) which are just used for training. Test set contains data for each user. For each user, there are several input graphs (e.g., regular user authentication activity graph) and several target graphs (e.g., malware user authentication activity graph). Input and target graphs in test set are not one-to-one, which can be tested by indirect evaluation. There are no node attributes for this dataset, and only edge attribute is considered. For each graph, the value of the $i - th$ row and the $j - th$ column refers to the edge attribute of node $i$ and $j$ (0 refers to no links). This dataset can be employed for weighted graph generation.

### C.7.1 License

**AuthNet**: The dataset is publically released by LANL [97]. To the extent possible under law, LANL has waived all copyright and related or neighboring rights to User-Computer Authentication Associations in Time. This work is published from: United States.

We collect this dataset from DBLP-Citation-network V12. We chose authors with affiliations, papers with more than one authors, and the time period from 1990 to 2019. To cut the graphs into pieces, we generate sub-graphs based on the fields of study of papers. For each field, we generate one spatio-temporal graph. Then we concatenate and pad all graphs, and save them into Numpy arrays. We save the graphs in .npy format.

### C.8 IoT Networks

**IoTNet** is the malware dataset collected for malware confinement prediction [31]. There are three sets of IoT nodes at different amounts (20, 40 and 60) encompassing temperature sensors connected with Intel ATLASEDGE Board and Beagle Boards (BeagleBone Blue), communicating via Bluetooth protocol. Benign and malware activities are executed on these devices to generate the initial attacked networks (i.e., the Internet of Things) as input graphs. Benign activities include MiBench [98] and SPEC2006 [99], Linux system programs, and word processors. The nodes represent devices and node attribute is a binary value referring to whether the device is compromised or not. Edge represents the connection of two devices and the edge attribute is a continuous value reflecting the distance of two devices. The real target graphs are generated by the classical malware confinement method: stochastic controlling with malware detection [100, 101, 102]. We collect 334 pairs of input and target graphs with different contextual parameters (infection rate, recovery rate and decay rate) for each of the three datasets. In this dataset, there are both nodes attributes and edge attributes considered. IoTNet can be used for attributed graph generation and weighted graph generation.

The original format of IoTNet contains 1,029 .csv files, we convert them to .npy files, input_adj.npy, input_edge.npy, input_node.npy, target_adj.npy, target_edge.npy, target_node.npy, Iot_20_labels.npy, Iot_40_labels.npy and Iot_60_labels.npy, to contain structure, node features, edge features and labels and to be easily read by Python. The detailed information of the data can be found in the corresponding readme.txt file. To reformat the data, we use glob to read in all .csv files from the directory, and separate the original .csv files into input data and target data; For both input and target data, we get edge feature from the original .csv files, get node feature(0 or 1 for IoTNet) from the diagonals of each file, and get adjacent matrix from the edge feature while setting the diagonals to be 0. For IoTNet, we also split the name and get labels from the name of each .csv file. We then reshaped all arrays into the required dimensions and converted them to NumPy files.

### C.8.1 License

**IoTNet**: The dataset is generated by [31]. The license is not specified.

### C.9 Skeleton Graphs

**Kinetics** [51] dataset is a large-scale human action dataset with 300000 videos clips in 400 classes. Those video clips are from YouTube with a great variety. The raw Kinetics dataset doesn't contain skeleton data, and [51] uses OpenPose toolbox to generate skeleton with 18 joints on every frame. Kinetics-Skeleton contains 240000 clips of training data and 20000 clips of test data. This dataset does not contain node or edge attributes, but contain temporal and 2D spatial features to be used in spatial graph generation and temporal graph generation tasks.

The raw Kinetics dataset is stored in a few .json files, and each json file contains information of a single video clip. We traverse all .json files, and concatenate their contents into several Numpy arrays with paddings for short video clips. We then remove extra skeletons, and leave each video clip only one skeleton. Finally, we save the data in .npy array.

**NTU-RGB+D** [52] dataset is a large and widely used action recognition dataset with 56000 action clips in 60 classes. These clips are performed by 40 volunteers captured in a constrained lab environment, with three camera views recorded simultaneously. The dataset provides 3D joint

locations of each frame and 25 joints for each subject. NTU-RGB+D does not contain node or edge attributes, but contain temporal and 3D spatial features to be used in 3D spatial graph generation and temporal graph generation tasks.

We process this dataset by the code from github. The dataset is originally stored in a few files, and each contains information of one single video clip. After the same processing process as we do for Kinetics dataset, we save the data in .npy format.

### C.9.1   License

**Skeleton (Kinectics)**: CC BY 4.0.

**Skeleton (NTU-RGB+D)**: Not specified.

### C.10   Social Networks

**Ego**: Ego dataset contains 757 3-hop ego networks extracted from the Citeseer [103]. The number of nodes of the graph in Ego dataset ranges from 50 to 399, and 145 in average. Each graph in Edo has around 335 edges. Nodes represent documents and edges represent citation relationships [34]. Ego does not contain node or edge attributes, and can be used for graph generation tasks.

**TwitterNet**: The dataset is processed by [56] and obtained from 5 different countries in Latin America, namely Brazil, Colombia, Mexico, Paraguay, and Venezuela. Data sources from Twitter are adopted as the model inputs. In each case the data for the period from July 1, 2013 to February 9, 2014 is used for training and validation, where the validation set consists of a randomly chosen $30\%$ of the data, and the rest is used for training; the data from February 10, 2014 to December 31, 2014 is used for the performance evaluation. TwitterNet contains 2,580 graphs in total, each of which has 300 nodes and 0.5 edges in average. This dataset can be employed in graph transformation tasks.

### C.10.1   License

**Ego**: This dataset is extracted from Citeseer [103]. Citeserr is under CC BY-NC-SA 3.0.

**TwitterNet**: The dataset is obtained from [104]. The license is not specified.

### C.11   Scene Graphs

**CLEVR** [50] dataset provides a dataset for visual question answer, which can be formalized as a spatial-graph dataset. CLEVR dataset contains 85,000 graphs in total. There are 10 objects in the image with different 3D locations. Each object is identified by its shape, such as sphere, cylinder, and cube. The relationship between two objects can be categorized into four types: right, behind, front, left, with directions. Thus, each image can be formalized as a labeled directed graph with different edge types and node types. Thus, the spatial information of each nodes is closely correlated with the edge types between each pair of nodes. As a result, CLEVR dataset can be employed for attributed graph generation, weighted graph generation and spatial graph generation.

### C.11.1   License

**CLEVR**: CC BY 4.0.

### C.12   Synthetic Graphs

**Barab'asi-Albert Graphs**: This dataset is generated by the Barab'asi-Albert model [31]. It fits the "one-to-one" mapping problem of graph translation. It contains pairs of input and target graphs. The target graph topology is the 2-hop connection of the input graph, where each edge in the target graph refers to the 3-hop reachability in the input graph (e.g., if node $i$ is 3-hop reachable to node $j$ in the input graph, then they are connected in the target graph). There are edge and node attributes for graphs in this dataset: the edge attribute $E_{(i,j)}$ denotes the existence of the edge, and the node attributes are continuous values computed following the polynomial function: $f(x) : y = ax^2 + bx + c$ $(a = 0; b = 1; c = 5)$, where $x$ is the node degree and $f(x)$ is the node attribute. Here we provide the datasets with three different node sizes. Barab'asi-Albert Graphs dataset can be used for attributed graph transformation.

The original Barab'asi-Albert Graphs dataset contains 3,000 .csv files. We reformat them into .npy files, including input_adj.npy, input_edge.npy, input_node.npy, target_adj.npy and target_edge.npy, target_node.npy for the community to use. To reformat the data, we use glob to read in all .csv files from the directory, and separate the original .csv files into input data and target data; For both input and target data, we get edge feature from the original .csv files, get node feature from the diagonals of each file, and get adjacent matrix from the edge feature while setting the diagonals to be 0. We then reshaped all arrays into the required dimensions and converted them to NumPy files.

**Community**: This dataset is generate by [34] and contains 3,000 two-community graphs, each of which has 64 nodes and around 340 edges. Each community is generated by the Erdos-Renyi model (E-R) [105] with $\frac{|V|}{2}$ nodes and the edge probability of 0.3. Then add $0.05|V|$ inter-community edges are added with uniform probability. This dataset does not have node or edge attributes. Community can be used for graph generation tasks.

**Erdos-Renyi Graphs**: This dataset is generated by the Erdos-Renyi model with the edge probability of 0.2 [31]. It fits the "one-to-one" mapping problem of graph translation. It contains pairs of (input, target) graphs. The target graph topology is the 2-hop connection of the input graph, where each edge in the target graph refers to the 2-hop reachability in the input graph (e.g., if node $i$ is 2-hop reachable to node $j$ in the input graph, then they are connected in the target graph). There are edge and node attributes for graphs in this dataset: the edge attribute $E_{(i,j)}$ denotes the existence of the edge, and node attributes are continuous values computed following the polynomial function: $f(x) : y = ax^2 + bx + c$ ($a = 0; b = 1; c = 5$), where $x$ is the node degree and $f(x)$ is the node attribute. This dataset contains 1,000 graphs in total, and can be used for attributed graph generation. The original Erdos-Renyi Graphs dataset contains 3,000 .csv files. We reformat them into .npy files, including input_adj.npy, input_edge.npy, input_node.npy, target_adj.npy and target_edge.npy, target_node.npy for the community to use. Detailed information can be found in ER_Readme.trf. To reformat the data, we use glob to read in all .csv files from the directory, and separate the original .csv files into input data and target data; For both input and target data, we get edge feature from the original .csv files, get node feature from the diagonals of each file, and get adjacent matrix from the edge feature while setting the diagonals to be 0. We then reshaped all arrays into the required dimensions and converted them to NumPy files.

**Scale-free**: This dataset is generated as a directed scale-free network [41], which is a network whose degree distribution follows power-law property [83]. It fits the "one-to-many" mapping graph translation problem. There are no node features in this dataset, and the goal is to learn the mapping from the input graph's topology to the target graph's topology. To generate a target graph, a node will by selected as target node with probability proportional to its in-degree, which will be linked to a new source node with probability of 0.41. Similarly, a node will be selected as the source node with the probability proportional to its out-degree, which will be linked to a new target node with the probability of 0.54. Then, a corresponding target graph is generated by adding m (number of nodes of the input graph) edges between two nodes. Thus, both input and target graphs are directed scale-free graphs. This dataset contains 10,000 graphs in total, and can be splitted into subsets that contains 10, 20, 50, 100, 150 nodes along with 20, 40, 100, 200 and 320 edges, respectively. The original Scale-free dataset contains 10,000 .csv files and we convert it to .npy files for people to read in Python. The detailed information of the data can be found in the corresponding scale_free_Readne.rtf. To reformat the data, we use glob to read in all .csv files from the directory, and separate the original .csv files into input data and target data; For both input and target data, we get edge feature from the original .csv files, get node feature from the diagonals of each file, and get adjacent matrix from the edge feature while setting the diagonals to be 0. Due to the massive .csv files in the Scale-free Graphs, we optimize to reduce the time complexity in order to process the dataset faster. We then reshaped all arrays into the required dimensions and converted them to NumPy files.

**Waxman Graphs**: This datase contains graphs generated by the Waxman random graph model that places $n$ nodes uniformly at random in a rectangular domain [106, 29]. There are three types of factors that are related to the generation of Waxman graphs: the independent graph factor $b$ that controls node attributes, the independent spatial factor $p$ that controls the overall node positions, and the graph-spatial correlated factor $s$ that controls both graph and spatial density [29]. There are 80,000 samples for training and 80,000 for testing. Each graph in the dataset contains 25 nodes and around 250 edges. Waxman Graphs dataset can be used for a few tasks, including attributed graph generation, spatial graph generation and temporal graph generation. The original Waxman Graphs dataset contains 96,000 graph files saved in Numpy array. We reformat them into .npy files, including adj.npy, edge_feat.npy, label.npy, node_feat.npy, spatial.npy, temporal_adj.npy, temporal_edge.npy, temporal_label.npy, temporal_node.npy and temporal_spatial.npy. The detailed informatiom can be found in waxman_Readme.rtf. To reformat these files, we load the testing and training dataset and converted the sparse matrices to dense matrices. we concatenate the testing and training datasets and reshape them into the required dimensions. To get the version of datasets with temporal dimension, we flattened the NumPy arrays. All datasets were saved as NumPy files eventually.

**Random Geometric Graphs**: This datase contains graphs generated by the random geometric graph model that places $n$ nodes uniformly at random in a rectangular domain [29]. Two nodes are joined by an edge if their distance is larger than a threshold $\beta = 12$. The node attributes among a graph are generated in the same rule as that for generating Waxman graphs. There are 8,000 samples for training and 1,600 for testing in this dataset. Each graph in the dataset contains 25 nodes and around 350 edges. Random Geometric Graphs dataset can be used for a few tasks, including attributed graph generation, spatial graph generation and temporal graph generation.

The original Random Geometric Graphs dataset contains 96,000 graph files saved in Numpy array. We reformat them into .npy files, including adj.npy, edge_feat.npy, label.npy, node_feat.npy, spatial.npy, temporal_adj.npy, temporal_edge.npy, temporal_label.npy, temporal_node.npy and temporal_spatial.npy. The detailed informatiom can be found in random_geo_Readme.rtf. To reformat these files, we load the testing and training dataset and converted the sparse matrices to dense matrices. we concatenate the testing and training datasets and reshape them into the required dimensions. To get the version of datasets with temporal dimension, we flattened the NumPy arrays. All datasets were saved as NumPy files eventually.

### C.12.1 License

**Barab'asi-Albert Graphs**: The dataset is generated by [31]. The license is not specified.

**Community**: The dataset is generated by [34], which is under MIT License. The license of the dataset is not specified.

**Erdos-Renyi graphs**: The dataset is generated by [31]. The license is not specified.

**Scale-free**: The dataset is generated by [41]. The license is not specified.

**Waxman graphs**: The dataset is generated by [29]. The license is not specified.

**Random geometric**: The dataset is generated by [29]. The license is not specified.

## D  Benchmark Results

We benchmark all the datasets with graph generation and transformation models. For graph generation task, we conduct experiments on three models, GraphRNN [34], GraphVAE [18], GraphGMG [8]. For graph transformation task, we conduct experiments on two models, Interaction Networks [38] and NEC-DGT [31].

### D.1  Molecule Generation Results.

As mentioned above, graph generation task could be very domain-specific, meaning that each domain has specific expectations over the generative tasks. Our first benchmark focuses on one of the most developed areas, molecular graph generation, which is motivated by drug and material discovery. For molecule generation task, we utilize the above mentioned self-quality based evaluation, where the validity, uniqueness and novelty are measured. We survey a list of state-of-the-art deep generative models on molecules and report the performance regarding validity, novelty, and uniqueness on two popular benchmark datasets (QM9 [44] and ZINC250K [45]) in the original paper as shown in Table 4. In Table 4, it is clearly to observe that the state-of-the-art models, such as MoFlow, GraphEBM, GraphDF, almost perform perfectly on the two common benchmarked datasets. As described in the following section, one key point to generate good molecular graphs is to handle the valency constraints. Some models utilize sequential generation, some utilize valency check, some design regularization, but overall, the best-performing models handle the valency constraint properly. However, it is not the end of the area. The molecule space being searched currently is small with very limited set of atoms and bonds and small size of molecules. Thus, benchmark datasets with larger molecules and molecules with more diverse atom and bond types are urgent to advance the field. From another perspective, it is important to generate molecules with desired properties which more domain-specific analyses and explorations could be done.

### D.2  Baseline Models

**GraphRNN [34].** GraphRNN represents graph generation as an auto-regressive process and builds an generative RNN model to generate nodes and edges sequentially.

**GraphVAE [18].** GraphVAE represents each graph by its adjacency matrix and feature vectors and utilizes graph neural network to encode the graphs into a vector space. Then, the model learns the distribution of the graphs via a VAE setting which minimizes the distance between the latent distribution and Gaussian distribution. Finally, the model decodes the latent vectors to reconstruct graphs.

**Table 4:** Quantitative evaluation and comparison on molecular graph generation tasks by different deep generative models on graphs ("Valid." is short for validness. "Novel." is short for novelty. "Unique." is short for uniqueness.).

| Method→ | QM9 | | | ZINC250K | | |
|---|---|---|---|---|---|---|
| Dataset↓ | Valid. | Novel. | Unique. | Valid. | Novel. | Unique. |
| GrammarVAE [107] | 31.00% | **100.00%** | 10.76% | 30.00% | 95.44% | 9.30% |
| GraphVAE [18] | 14.00% | **100.00%** | 31.60% | 61.00% | 85.00% | 40.90% |
| CGVAE [108] | **100.00%** | **100.00%** | 99.82% | **100.00%** | 94.35% | 98.54% |
| GraphNVP[86] | 74.30% | **100.00%** | 94.80% | 90.10% | 54.00% | 97.30% |
| GRF [109] | 73.40% | **100.00%** | 53.70% | 84.50% | 58.60% | 66.00% |
| GraphAF[35] | **100.00%** | **100.00%** | 99.10% | **100.00%** | 88.83% | 94.51% |
| CGSVAE [110] | 34.90% | **100.00%** | - | 96.60% | 97.50% | - |
| JT-VAE [20] | **100.00%** | **100.00%** | 99.80% | - | - | - |
| GCPN [93] | **100.00%** | **100.00%** | **99.97%** | - | - | - |
| MolecularRNN [36] | **100.00%** | **100.00%** | 99.89% | - | - | - |
| MolGAN [19] | - | - | - | 98.10% | 94.10% | 10.40% |
| MPGVAE [111] | - | - | - | 91.00% | 54.00% | 68.00% |
| SCAT [112] | - | - | - | 47.40% | 92.00% | 98.30% |
| MoFlow [32] | **100.00%** | 98.03% | 99.20% | **100.00%** | **100.00%** | **99.99%** |
| GraphEBM [33] | **100.00%** | 97.01% | 97.90% | 99.96% | **100.00%** | 98.79% |
| GraphDF [37] | **100.00%** | 98.10% | 97.62% | **100.00%** | **100.00%** | 99.55% |

**GraphGMG [8].** GraphGMG first learns a node-level embedding of a given graph, then learns a probability distribution over possible outcomes for each generation step. During the generation process, the model sequentially connects nodes and edges to a new graph.

**GrammarVAE [107].** GrammarVAE is one of the first deep generative models that learn to generative novel molecules with a string representation.

**GraphVAE [18].** GraphVAE is a VAE-based graph generative models that generates graphs in an one-shot fashion.

**CGVAE [108].** CGVAE is a VAE-based graph generative model that formulates the generation process as an iterative process.

**GraphNVP [86].** GraphNVP first introduces the idea of invertible normalizing flow-based methods to molecular graph generation in an one-shot generation way.

**GRF [109].** GRF introduces residual flows for molecular graph generation which circumvents the requirement of partitioning of the latent vector in GraphNVP.

**GraphAF [35].** GraphAF takes one step further than GraphNVP to formulate the problem as a sequential generation problem.

**CGSVAE [110].** CGSVAE is a VAE-based graph generative models that proposes a regularization method that encourages the model to generate valid molecules.

**JT-VAE [20].** JT-VAE is motivated to explicitly model substructures in the generative models that introduces an extra junction tree encoder-decoder part which each node denotes a substructure rather than an atom in a molecule.

**GCPN [93].** GCPN formulates molecular graph generation as a reinforcement learning problem where each state is a generation step, every step, it takes the action to connect two atoms and labels the edges by bond types. It stops when no atoms are connected.

**MolecularRNN [36].** MolecularRNN follows the idea of GraphRNN and adopts it for the molecular graph generation task.

**MolGAN [19].** MolGAN is a GAN-based molecular graph generation method that implements a GAN model to generate molecular graphs in an one-shot fashion.

**MPGVAE [111].** MPGVAE designs a VAE-based which follows Graphite [113] and generate molecular graphs in an one-shot way.

**SCAT [112].** SCAT takes a scattering transform and gaussianization as an encoder and utilizes a MLP as a decoder to generate novel molecular graphs in an one-shot way.

**MoFlow [32].** MoFlow improves over GraphNVP by introducing a valency correction mechanism in the framework.

Table 5: Hyper-parameters for graph generation benchmark.

| Method | Learning Rate | Epoch | Batch Size | # graphs |
|--------|---------------|-------|------------|----------|
| GraphRNN | $3\times10^{-3}$ | 1,000 | 32 | 1,000 |
| GraphVAE | $1\times10^{-3}$ | 10 | 1 | 1,000 |
| GraphGMG | $1\times10^{-3}$ | 10 | 1 | 1,000 |

Table 6: Hyper-parameters for graph transformation benchmark. Due the capacity of our memory, for the graph transformation task, we sampled a subset from a few datasets for evaluation. The size of the subset depends on the graph size and total number of graphs contained in the dataset.

| Dataset | Interaction Networks | | | NEC-DGT | | |
|---------|---------------|-------|---------|---------------|-------|---------|
| | Learning Rate | Epoch | #graphs | Learning Rate | Epoch | #graphs |
| AuthNet | $1\times10^{-2}$ | 100 | 412 | $1\times10^{-4}$ | 500 | 412 |
| Barab'asi-Albert Graphs | $1\times10^{-2}$ | 100 | 1,000 | $1\times10^{-4}$ | 500 | 1,000 |
| Brain-restingstate | $1\times10^{-2}$ | 100 | 823 | $1\times10^{-4}$ | 500 | 823 |
| Brain-emotion | $1\times10^{-2}$ | 100 | 811 | $1\times10^{-4}$ | 500 | 811 |
| Brain-grambling | $1\times10^{-2}$ | 100 | 818 | $1\times10^{-4}$ | 500 | 818 |
| Brain-language | $1\times10^{-2}$ | 100 | 816 | $1\times10^{-4}$ | 500 | 816 |
| Brain-motor | $1\times10^{-2}$ | 100 | 816 | $1\times10^{-4}$ | 500 | 816 |
| Brain-relational | $1\times10^{-2}$ | 100 | 808 | $1\times10^{-4}$ | 500 | 808 |
| Brain-social | $1\times10^{-2}$ | 100 | 816 | $1\times10^{-4}$ | 500 | 816 |
| Brain-wm | $1\times10^{-2}$ | 100 | 812 | $1\times10^{-4}$ | 500 | 812 |
| Scale-free | $1\times10^{-2}$ | 100 | 250 | $1\times10^{-4}$ | 500 | 250 |
| TwitterNet | $1\times10^{-2}$ | 100 | 250 | $1\times10^{-4}$ | 500 | 250 |
| N-body-charged | $1\times10^{-2}$ | 100 | 150 | $1\times10^{-4}$ | 500 | 150 |
| N-body-spring | $1\times10^{-2}$ | 100 | 150 | $1\times10^{-4}$ | 500 | 150 |
| ChemReact | $1\times10^{-2}$ | 100 | 1,000 | $1\times10^{-4}$ | 500 | 1,000 |
| IoTNet | $1\times10^{-2}$ | 100 | 343 | $1\times10^{-4}$ | 500 | 343 |
| MolOpt | $1\times10^{-2}$ | 100 | 500 | $1\times10^{-4}$ | 500 | 500 |

**GraphEBM [33].** GraphEBM is an energy-based generative model that utilizes Langevin Dynamics to sample novel molecules.

**GraphDF [37].** GraphDF improves over GraphAF by learning discrete latent variables rather than continuous latent variables as in most of the Flow and VAE-based methods.

**Interaction Network [38].** Physical domain is the target for Interaction Networks, the input of which is a graph that represents a system of objects and relations. Interaction Networks instantiates the pairwise interaction and compute its effects via a relational model. The effects are then aggregated and combined with the objects and external effects to generate the input for an object model, which predicts how the interactions and dynamics influence the objects.

**NEC-DGT [31].** In NEC-DGT, the node and edge attributes of input graphs are inputted to the model. The model outputs node attributes and edges attributes of the generated target graphs via several blocks, which have edge and node translation paths co-evolved and combined by a graph regularization during training process.

### D.3 Hyper-parameters

All experiments are conducted on a 64-bit machine with a 6 core Intel CPU i9-9820X, 32GB RAM, and an NVIDIA GPU (GeForce RTX 2080ti, 1545MHz, 11GB GDDR6). The detailed hyper-parameters can be found in Table 5 and Table 6. For the molecular graph generation benchmark, we take experiment results from the original reports.

## E    Tutorials

We provide data processors, evaluators, as well as visualizers which simplify the pipeline for graph generation and transformation, as shown in Fig. 4, 5 and 6, respectively.

```python
import graphgt
```

```python
qm9_data_loader = graphgt.DataLoader(name='qm9', save_path='./', format='numpy')
```

```
Downloading node feature...
100%|████████████████████████████████████████| 31.1M/31.1M [00:03<00:00, 9
.13MiB/s]
Done!
Downloading edge feature...
100%|████████████████████████████████████████| 49.5M/49.5M [00:04<00:00, 1
0.1MiB/s]
Done!
Downloading spatial feature...
100%|████████████████████████████████████████| 35.8M/35.8M [00:03<00:00, 9
.19MiB/s]
Done!
Downloading adjacency matrix...
100%|████████████████████████████████████████| 49.5M/49.5M [00:04<00:00, 1
0.2MiB/s]
Done!
Downloading smiles string...
100%|████████████████████████████████████████| 15.0M/15.0M [00:02<00:00, 6
.97MiB/s]
Done!
```

```python
adj, node_feat, edge_feat, spatial, smile = qm9_data_loader.get_data()
```

**Figure 4:** Loading generation dataset.

```python
import graphgt
```

```python
ER20_data_loader = graphgt.DataLoader(name='ER_20', save_path='./', format='numpy')
```

```
Downloading input node feature...
100%|████████████████████████████████████████| 80.1k/80.1k [00:00<00:00,
230kiB/s]
Done!
Downloading input edge feature...
100%|████████████████████████████████████████| 1.60M/1.60M [00:01<00:00, 1
.51MiB/s]
Done!
Downloading input adjacency matrix...
100%|████████████████████████████████████████| 800k/800k [00:00<00:00,
902kiB/s]
Done!
Downloading target node feature...
100%|████████████████████████████████████████| 80.1k/80.1k [00:00<00:00, 1
.20MiB/s]
Done!
Downloading target edge feature...
100%|████████████████████████████████████████| 1.60M/1.60M [00:01<00:00, 1
.54MiB/s]
Done!
Downloading adjacency matrix...
100%|████████████████████████████████████████| 800k/800k [00:00<00:00,
883kiB/s]
Done!
```

```python
input_adj, input_node_feat, input_edge_feat, input_spatial, target_adj, target_node_feat, target_edge_feat, target_s
```

**Figure 5:** Loading transforamtion dataset.

```python
import graphgt
import numpy as np
```

```python
batch = 1000
x = np.random.rand(batch,1)
y_baseline = np.random.rand(batch,1)
y_pred = np.zeros((batch,1))

print('MMD baseline', graphgt.compute_mmd(x,y_baseline))
print('MMD prediction', graphgt.compute_mmd(x,y_pred))
print ('KLD', graphgt.compute_kld(x,y_baseline))
print ('EMB', graphgt.compute_emd(x,y_baseline))
```

```
MMD baseline 9.684740112247958e-05
MMD prediction 0.3751574658037742
KLD [0.51577211]
EMB 0.01009273634128826
```

**Figure 6:** Evaluation APIs.