# OpenReview forum: "GraphGT: Machine Learning Datasets for Graph Generation and Transformation"
_NeurIPS.cc/2021/Track/Datasets_and_Benchmarks/Round2 — NeurIPS 2021 Datasets and Benchmarks Track (Round 2)_

### Official Review · Reviewer_RhvT · 2021-09-14
**A nice collection of datasets for new tasks**

**Rating:** 7
**Confidence:** 4
**Correctness:** I do not have any concerns with the c…

**Strengths:**

1. The collected datasets are diverse in terms of domains and subjects;
2. The task formulations are clear with motivating examples;
3. The evaluation pipeline is carefully normalized;
4. A python library and website are provided for easy usage;

**Weaknesses:**

1. There're only two datasets benchmarked in the paper, looking forwards to see more and willing to increase my rating if authors do so;
2. The benchmarked datasets seem to be easy because most of the baselines got 100% on evaluation metrics;
3. It might be better to use figures to illustrate the task formulation for easy understanding;

**Additional Feedback:**

All my feedback is outlined in the above sections.

**Clarity:**

The paper is overall well written but could be more concise on the dataset descriptions.

**Documentation:**

The dataset collection/processing procedures are detailed. The datasets are well-organized and availabilities are clearly stated. The copyrights are enumerated for every dataset. All required materials are properly provided and explained.

I recommend also putting the documentations on the website under each dataset.

**Ethics:**

I do not have any concerns

**Relation To Prior Work:**

The paper clearly contextualizes its contributions over previous work.

**Summary And Contributions:**

This work presents a benchmark called GraphGT for graph generation and transformation tasks for the first time. The collected datasets cover various domains and subjects and are released with useable APIs. It also benchmarks two datasets with three evaluation metrics.

---

> ### Author Response · Authors · 2021-09-27
> **Author's response**
>
> We appreciate your valuable comments and feedback that help us improve the paper!
>
> Review #1: There're only two datasets benchmarked in the paper, looking forwards to see more and willing to increase my rating if authors do so;
>
> Response: Thanks for your comments.
> * We have added benchmark results for 16 graph generation datasets and 17 graph transformation datasets with popular graph generative models in section 5.1.2 (Line #342-360) and 5.2.2 (Line #386-408).
> * The detailed baseline models, hyper-parameters, experiment set-ups can be found in Appendix D (Line #1219-1277).
> * We also added the code to run the models on our GitHub Repo (https://github.com/yuanqidu/GraphGT).
>
>
> Review #2: The benchmarked datasets seem to be easy because most of the baselines got 100% on evaluation metrics;
>
> Response: Thanks for your comments. For the molecule generation problem, the three metrics (validity, novelty, uniqueness) are only part of the evaluations. There are more domain-specific evaluations (e.g. generating molecules with desired properties). Due to the page limit, we have moved it to Appendix D and added more analyses about the results and further evaluations (Line #1201-1218). Moreover, QM9 and ZINC250K are two commonly used datasets in this field and a huge amount of graph generative models were evaluated based on these two datasets. Their indistinguishable performance on several SOTA models indicates that more benchmark datasets are needed in this field, which is one of the motivations of GraphGT.
>
> Review #3: It might be better to use figures to illustrate the task formulation for easy understanding;
>
> Response: Thanks for your comments. We have added an illustrative figure (Figure 1) as a summary of GraphGT. Besides, we have also polished our Figure 2 and Figure 3.
>
> Review #4: The paper is overall well written but could be more concise on the dataset descriptions.
>
> Response: Thanks for your comments.
> * We have completely re-written the dataset details part (section 4.2, Line #167-303) with an organized format: motivation, task, data construction.
> * We elucidated motivations, added concrete examples, and related references for all domains covered by GraphGT.
> * We also provided the general procedures on the efforts we devoted when constructing the dataset, with even further details in appendix C (Line #783-1195).
>
> Review #5: I recommend also putting the documentations on the website under each dataset.
>
> Response: Thanks for your suggestion. We have uploaded all the documents to our website. Our website is kindly attached again https://graphgt.github.io/

---

> > ### Comment · Reviewer_RhvT · 2021-09-27
> > **Thank you for your response**
> >
> > Hi,
> >
> > Thank you for your response. I think the new experiments are good and the revised paper addressed my concern. Thus I raised my score.

---

### Official Review · Reviewer_3P8C · 2021-09-16
**Technically sound; rooms for improvements**

**Rating:** 7
**Confidence:** 4
**Correctness:** No technical flaws are spotted.

**Strengths:**

+ Graph generation and transformation are two important topics for many real-world problems.

+ A handy library featuring dataset loading and evaluation is provided.

+ Experiments on state-of-the-art methods of graph generation with various evaluation metrics are conducted.



**Weaknesses:**

- Experiments are limited to one task, i.e. chemical molecular graph generation, only.

- More discussions should be given to better illustrate the background, real-world impact, challenges, and status of existing studies. See my detailed comments below.

- The development of graph generation and transformation models should be outlined in the related work part.


**Additional Feedback:**

1. As the authors point out in the main text, the problem of graph generation and transformation may differ in contexts and domains. Therefore, it is suggested to compare the implementations of graph generation and transformation in different domains, so as to allow readers to understand the limitations and difficulties. In the current version, some baselines methods target at specific domains, e.g., MoFlow for molecule graph generation, while some others are for general domains, e.g., GraphRNN. A comparative study may be helpful for better showing the difference in different domains.

2. Deeper discussions on the experimental results should be provided, in particular, how do existing work differs from each other and how different metrics affect model performance. It is interesting for me to discuss the choice of evaluation metrics, which seems are most based on simple statistics and heuristics. From Table 2, it is observed that many existing baselines have already achieved quite promising performance in terms of the concerned three metrics; does it mean that the generation quality is good enough, or may other domain-specific metrics be adopted? The authors may also outline several future research directions that allow for improvements.

3. Minor edits:
  - In Line 322, what is "label graph"?
  - In Line 352, "Fig. 2" should be Table 2.

**Clarity:**

The paper reads somewhat quirky but is easy to follow overall. More background knowledge of real-world applications may better demonstrate the impact of this problem.


**Documentation:**

All datasets are clearly documented in the hosted website. Some links to details of the problems are not accessible.

**Relation To Prior Work:**

The relation to prior datasets in relevant fields have been examined in the related work section.

**Summary And Contributions:**

This paper systematically studies the problem of graph generation and transformation. Specifically, 36 datasets from various domains are collected and organized into a handy library; benchmarking studies on generation for chemical datasets are conducted to demonstrate the effectiveness of existing state-of-the-arts.

---

> ### Author Response · Authors · 2021-09-27
> **Author's response (1/2)**
>
> We appreciate your valuable comments and feedback that help us improve the paper!
>
> Review #1: Experiments are limited to one task, i.e. chemical molecular graph generation, only.
>
> Response: Thanks for your suggestion.
> * We have added graph generation benchmark results and graph transformation benchmark results and analyses for 16 graph generation datasets and 17 graph transformation datasets with popular graph generation models in section 5.1.2 (Line #342-360) and section 5.2.2 (Line #386-408),
> * The experiment details can be found in Appendix D (Line #1219-1277) including baseline models, hyper-parameters, experiment set-ups.
> * We also added the code to run the models on our GitHub Repo (https://github.com/yuanqidu/GraphGT).
>
> Review #2: More discussions should be given to better illustrate the background, real-world impact, challenges, and status of existing studies. See my detailed comments below.
>
> Response: Thanks for your comments.
> * We have completely re-written the dataset details part (section 4.2, Line #167-303) with an organized format: motivation, task, data construction.
> * We elucidated motivations, added concrete examples, and related references for all domains covered by GraphGT.
> * We also provided the general procedures on the efforts we devoted when constructing the dataset, with even further details in appendix C (Line #783-1195).
>
>
> Review #3: The development of graph generation and transformation models should be outlined in the related work part.
>
> Response: Thanks for your comments. We have added the related works right after the problem formulation in sections 3.1 (Line #129-134) and 3.2 (Line #149-156).
>
> Review #4: The paper reads somewhat quirky but is easy to follow overall. More background knowledge of real-world applications may better demonstrate the impact of this problem.
>
> Response: Thanks for your comments. We have rewritten section 4.2 for each domain to include motivations, tasks in a deeper and more organized way (Line #167-303).
>
> Review #5: All datasets are clearly documented in the hosted website. Some links to details of the problems are not accessible.
>
> Response: Thanks for your comments. We have fixed the problem and we will continue to include more materials, such as new datasets, evaluation functions, tutorials, workshops, lectures, and surveys in the future. Our website is kindly attached again https://graphgt.github.io/
>
> Review #6: As the authors point out in the main text, the problem of graph generation and transformation may differ in contexts and domains. Therefore, it is suggested to compare the implementations of graph generation and transformation in different domains, so as to allow readers to understand the limitations and difficulties. In the current version, some baselines methods target at specific domains, e.g., MoFlow for molecule graph generation, while some others are for general domains, e.g., GraphRNN. A comparative study may be helpful for better showing the difference in different domains.
>
> Response: Thanks for your comments.
> * We have added graph generation benchmark results and graph transformation benchmark results and analyses for 16 graph generation datasets and 17 graph transformation datasets with popular graph generation models in section 5.1.2 (Line #342-360) and section 5.2.2 (Line #386-408).
> * We have analyzed the results and found that domains that existing comparison methods typically perform better on datasets with small graphs than those with larger graphs. Domains that have datasets corresponding to complex systems, such as brain network datasets, often challenge the relatively simple models more, such as Interaction Networks, and seem to need more advanced models to achieve better performance. The performance is also influenced by model properties, such as what types of features (e.g., spatial, ordinal, categorical, etc.) it can handle. The above patterns have not been observed in the previous papers of the comparison methods which only employed limited number of datasets in very focused domains. Therefore, this further highlights that our large number of datasets can help provide much more comprehensive evaluations and comparisons for comparison methods in graph generation. More importantly, it can help pinpoint the drawbacks of existing methods that are difficult to be discovered in existing few datasets which might have already been “overfit” by many existing methods. Hence, we believe our GraphGT is very helpful in overcoming the current bottleneck of generation and transformation method domain and is highly important in advancing this promising area.
> * We have highlighted the problems and future directions in some well-developed domains, e.g. molecules in section D.1 (Line #1201-1218).
> * We have added more discussion about related works in section 3.1 (Line #129-134) and 3.2 (Line #149-156) and also in Appendix D.2 (Line #1219-1272), we gave a more detailed introduction about each comparison method.

---

> ### Author Response · Authors · 2021-09-27
> **Author's response (2/2)**
>
> Review #7: Deeper discussions on the experimental results should be provided, in particular, how do existing work differs from each other and how different metrics affect model performance. It is interesting for me to discuss the choice of evaluation metrics, which seems are most based on simple statistics and heuristics. From Table 2, it is observed that many existing baselines have already achieved quite promising performance in terms of the concerned three metrics; does it mean that the generation quality is good enough, or may other domain-specific metrics be adopted? The authors may also outline several future research directions that allow for improvements.
>
> Response: Thanks for your comments!
> * We have added deeper discussions about the benchmark results in section 5.1.2 (Line #342-360) and 5.2.2 (Line #386-408) as well as appendix D.1 (Line #1201-1218).
> * We have discussed the details about evaluation metrics in section 5.1.1 (Line #306-341) and 5.2.1 (Line #362-385) for graph generation and transformation tasks
> * We have discussed the experiment results, analyzed why they achieved quite good results in terms of the metrics, further evaluations, and future directions to advance the field.
>
> Review #8: Minor edits:In Line 322, what is "label graph"? In Line 352, "Fig. 2" should be Table 2.
>
> Response: Thanks for carefully reading our paper, we have corrected the “label graph” to “target graph” and “Fig” to “Table”, and carefully proofread the paper again.

---

> ### Comment · Reviewer_3P8C · 2021-09-28
> **Reviewer's Response**
>
> Thanks for detailed response. I appreciate the substantial amount of revision the authors made and I would like to support the acceptance of this work.

---

### Official Review · Reviewer_PsXw · 2021-09-17
**More like a summarisation of datasets in previous works with limited contribution for graph generation and transformation**

**Rating:** 6
**Confidence:** 5

**Strengths:**

* A vast range of domains are considered, including biology, engineering, social science, etc.
* Different tasks of graph generation and graph transformation are covered by the collected datasets.
* Various metrics are provided to evaluate graph generation and transformation.

**Weaknesses:**

* It seems that all datasets are collected from previous works (Table 1), there is no newly built dataset. It is just a simple summarization of datasets in previous works. There is no discussion on why these datasets are chosen and how significant they are for graph generation and transformation. It remains unclear what kind of datasets are really useful for these tasks or any criteria when building related datasets.
* The detailed description of all datasets takes a major part of the paper (which also appears in the Appendix), however, they are all from existing works which do not cover the contribution of this work.
* The benchmark part is too thin, without any detail about the considered methods or experiment setup (even in appendix), which makes the results less convincing. There isn’t any analysis on the results, e.g. how do they differ from or agree with results in previous works?
* Overall, this paper is more like a summarization of datasets in previous works with limited contribution for future research in graph generation and transformation.

**Additional Feedback:**

My suggestion to the authors is to think first what is actually the challenges in graph generation and transformation. A simple summarization of datasets in previous works doesn't make much sense for the community. It would be more important to find out how datasets or benchmarks can cover limitations in previous works of graph generation and transformation, and contribute to further research in this domain.

**Clarity:**

There are several obvious errors. In line 111, the dimension of edges is N, equal to the dimension of nodes, which is obviously wrong. In line 135, there is a clear error in the vector space of the target domain.

**Correctness:**

All datasets are collected from previous works. Some of them are processed, but none of them are entirely new. The benchmark part is poorly represented without any detail of experiments and code, thus the results are not guaranteed to be correct.

**Documentation:**

Details of datasets are provided, as well as the availability and maintenance.

**Ethics:**

No ethical concerns.

**Relation To Prior Work:**

The authors only discuss datasets in other domains like node classification and discuss that they are not appropriate for graph generation and transformation. However, there is no discussion about how the collected datasets differ from those used in previous works of graph generation and transformation.

**Summary And Contributions:**

This paper proposes GraphGT, containing 36 datasets from 9 domains across 6 subjects, which assists the research on graph generation and graph transformation tasks. The authors provide a systematic review and classification of the datasets and provide APIs for the graph generation pipeline, which simplifies data loading, experimental setup, and evaluation.

---

> ### Author Response · Authors · 2021-09-27
> **Author's response (1/3)**
>
> We appreciate your valuable comments and feedback that help us improve the paper!
>
> Review #1: It seems that all datasets are collected from previous works (Table 1), there is no newly built dataset. It is just a simple summarization of datasets in previous works. There is no discussion on why these datasets are chosen and how significant they are for graph generation and transformation. It remains unclear what kind of datasets are really useful for these tasks or any criteria when building related datasets.
>
> Response: Thanks for your comments.
> * We have significant contributions in dataset construction. For example, CollabNet dataset and 7 brain network datasets are collected by us and constructed from scratch for graph generation and transformation. Another 8 datasets are re-purposed by us from other application into graph generation and transformation tasks for the first time. The remaining are from very different domains that share quite different terminology, format, and data structure, which are reformatted by us to a unified format for the first time for easy access and use in a standardized manner. The detailed information of how we collect, re-purpose, reformat each of the dataset can be found in Appendix C (Line #783-1195). We also explicitly added data construction in section 4.2 (Line #167-303).
> * Graph generation and transformation in machine learning are significant and promising in the domains we raised up in the paper. It is well-recognized that network generative models in transportation [3], biology [4] and physics [2] domains are significant, as summarized in many classical network science books and surveys such as [1] and [5]. In recent years, a fast-increasing number of machine learning models (e.g., deep graph generative models) have been proposed which need massive data for training and evaluation, unlike limited number of datasets evaluated by traditional network science models. Therefore, we collected and repurposed a large amount of data to this task. We have clarified this in section 1 (#Line 45-87) and Conclusion section (#Line409-421).
> * The datasets that are useful for our tasks should be those that contain a large number of networks, for sufficiently training machine learning models. For graph generation tasks, a large number of networks are used to learn the graph generative models that can fit the distributions of the graph data. For graph transformation tasks, a large number of pairs of source graphs and target graphs are needed in order to learn their (stochastic) mapping. The datasets that can satisfy these criteria will be considered as viable datasets. We have clarified this in section 1 (#Line 46-64, ), section 2 (#Line 105-111).
>
> [1] Newman, M. E. J. (2010). Networks: An introduction. Oxford: Oxford University Press.
>
> [2] Perraudin, N., Srivastava, A., Lucchi, A., Kacprzak, T., Hofmann, T., & Réfrégier, A. (2019). Cosmological N-body simulations: a challenge for scalable generative models. Computational Astrophysics and Cosmology, 6(1), 1-17.
>
> [3] Magnanti, T. L., & Wong, R. T. (1984). Network design and transportation planning: Models and algorithms. Transportation science, 18(1), 1-55.
>
> [4] Sanchez-Lengeling, B., & Aspuru-Guzik, A. (2018). Inverse molecular design using machine learning: Generative models for matter engineering. Science, 361(6400), 360-365.
>
> [5] Barthélemy, Marc. "Spatial networks." Physics Reports 499.1-3 (2011): 1-101.
>
> Review #2: The detailed description of all datasets takes a major part of the paper (which also appears in the Appendix), however, they are all from existing works which do not cover the contribution of this work.
>
> Response: Thanks for your comments. We have completely rewritten section 4.2 (Line #167-303) with a unified format of motivations, task descriptions and data construction. The more detailed information about how we processed each dataset and the dataset format can be found in Appendix C (Line #783-1195).

---

> ### Author Response · Authors · 2021-09-27
> **Author's response (2/3)**
>
> Review #3: The benchmark part is too thin, without any detail about the considered methods or experiment setup (even in appendix), which makes the results less convincing. There isn’t any analysis on the results, e.g. how do they differ from or agree with results in previous works?
>
> Response: Thanks for your comments and we have significantly extended the paper to address the issues mentioned.
> * We have added graph generation benchmark results and graph transformation benchmark results and analyses for 16 graph generation datasets and 17 graph transformation datasets with popular graph generation models in section 5.1.2 (Line #342-360) and section 5.2.2 (Line #386-408).
> * The experiment details can be found in Appendix D (Line #1219-1277) including baseline models, hyper-parameters, and experiment set-ups.
> * We also added the code to run the models on our GitHub Repo (https://github.com/yuanqidu/GraphGT).
> * We have analyzed the results and found that domains that existing comparison methods typically perform better on datasets with small graphs than those with larger graphs. Domains that have datasets corresponding to complex systems, such as brain network datasets, often challenge the relatively simple models more, such as Interaction Networks, and seem to need more advanced models to achieve better performance. The performance is also influenced by model properties, such as what types of features (e.g., spatial, ordinal, categorical, etc.) it can handle. The above patterns have not been observed in the previous papers of the comparison methods which only employed limited number of datasets in very focused domains. Therefore, this further highlights that our large number of datasets can help provide much more comprehensive evaluations and comparisons for comparison methods in graph generation. More importantly, it can help pinpoint the drawbacks of existing methods that are difficult to be discovered in existing few datasets which might have already been “overfit” by many existing methods. Hence, we believe our GraphGT is very helpful in overcoming the current bottleneck of generation and transformation method domain and is highly important in advancing this promising area.
>
>
> Review #4: Overall, this paper is more like a summarization of datasets in previous works with limited contribution for future research in graph generation and transformation.
>
> Response: Thanks for your comments. There are five main challenges that are deemed the bottlenecks against the advance of the graph generation and transformation field (We have added the clarification also in Section 1. Line #45-63):
> * Difficulty in formulation: graph structured data is complex in its nature; and the raw data in different domains may require greatly different procedures to process or re-process in order to fit into a unified format.
> * Limited number of application domains: Although graph generation and transformation is a very broad generic concept that covers graphs in areas ranging from geography to biology, to physics, to sociology, to engineering, existing datasets only cover limited domains which prevents the development of graph generative models as well as applications in more diverse domains.
> * Lack of taxonomy: As the area of graph generation and transformation grows, the research tasks are diversified and hence require a well-defined categorization in order to have the dataset under the right category for the evaluation of the corresponding task.
> * Lack of unified evaluation procedures: the evaluation metrics used in existing research works are quite diverse and a gold standard for the evaluation procedure and metrics is needed. Moreover, the scarcity of existing datasets may bias the selection of elevation metrics to fit the limited number of existing datasets (e.g., molecules) but may not be general to other datasets.
> * Lack of rigorous model comparisons: the model comparisons of existing work are only conducted on a small group of datasets, which may not necessarily reflect the performance of the models.
>
> Our work makes the following contributions to tackle the above challenges (In Section 1. Line #64-87):
>
> * We collect, re-purpose, re-format a large amount of graph datasets
> * GraphGT covers a variety of domains and subjects
> * GraphGT provides a systematic reviews and classifications of the datasets
> * GraphGT standardizes on the model evaluation procedures
> * GraphGT provides benchmark results on a large amount of datasets.

---

> ### Author Response · Authors · 2021-09-27
> **Author's response (3/3)**
>
> Review #5: All datasets are collected from previous works. Some of them are processed, but none of them are entirely new. The benchmark part is poorly represented without any detail of experiments and code, thus the results are not guaranteed to be correct.
>
> Response: Thanks for your comments.
> * Among all the datasets, CollabNet dataset and 7 brain network datasets were collected by us and constructed from scratch for graph generation and transformation. Another 8 datasets are re-purposed by us from other applications into graph generation and transformation tasks for the first time. The remaining are from very different domains that share quite different terminology, formats, and data structures, which are reformatted by us to a unified format for the first time for easy access and use in a standardized manner. In addition, the remaining datasets are from very different domains that share quite different terminology, format, and data structure, which are reformatted by us to a unified format for the first time for easy access and use in a standardized manner.
> * We have added graph generation benchmark results and graph transformation benchmark results and analyses for 16 graph generation datasets and 17 graph transformation datasets with popular graph generation models in section 5.1.2 (Line #342-360) and section 5.2.2 (Line #386-408).
> * The experiment details can be found in Appendix D (Line #1219-1277) including baseline models, hyper-parameters, experiment set-ups.
> * We also added the code to run the models on our GitHub Repo (https://github.com/yuanqidu/GraphGT).
>
> Review #6: There are several obvious errors. In line 111, the dimension of edges is N, equal to the dimension of nodes, which is obviously wrong. In line 135, there is a clear error in the vector space of the target domain.
>
> Response: Thanks for carefully reading our paper. We have fixed the errors and carefully proofread the paper again.
>
> Review #7: The authors only discuss datasets in other domains like node classification and discuss that they are not appropriate for graph generation and transformation. However, there is no discussion about how the collected datasets differ from those used in previous works of graph generation and transformation.
>
> Response: Thanks for your comments. We have added a more detailed discussion about previous works (graph generation datasets) in section 2 (Line #88-111) and re-stated our contribution to tackle the challenges in the graph generation and transformation field in section 1 (Line #64-87).
>
> Review #8: My suggestion to the authors is to think first what is actually the challenges in graph generation and transformation. A simple summarization of datasets in previous works doesn't make much sense for the community. It would be more important to find out how datasets or benchmarks can cover limitations in previous works of graph generation and transformation, and contribute to further research in this domain.
>
> Response: Thanks for your comments. We have identified 5 challenges in graph generation and transformation as follows (In Section 1, Line #45-63):
> * Difficulty in formulation: graph structured data is complex in its nature; and the raw data in different domains may requires greatly different procedures to process or re-process in order to fit into a unified format.
> * Limited number of application domains: Although graph generation and transformation is a very broad generic concept that covers graphs in areas ranging from geography to biology, to physics, to sociology, to engineering, existing datasets only cover limited domains which prevents the development of graph generative models as well as applications in more diverse domains.
> * Lack of taxonomy: As the area of graph generation and transformation grows, the research tasks are diversified and hence require a well-defined categorization in order to have the dataset under the right category for the evaluation of the corresponding task.
> * Lack of unified evaluation procedures: the evaluation metrics used in existing research works are quite diverse and a gold standard for the evaluation procedure and metrics is needed. Moreover, the scarcity of existing datasets may bias the selection of elevation metrics to fit the limited number of existing datasets (e.g., molecules) but may not be general to other datasets.
> * Lack of rigorous model comparisons: the model comparisons of existing work are only conducted on a small group of datasets, which may not necessarily reflect the performance of the models.
>
> Our work makes the following contributions to tackle the above challenges (In Section 1. Line #64-87):
> * We collect, re-purpose, re-format a large amount of graph datasets
> * GraphGT covers a variety of domains and subjects
> * GraphGT provides a systematic reviews and classifications of the datasets
> * GraphGT standardizes on the model evaluation procedures
> * GraphGT provides benchmark results on a large amount of datasets.

---

> ### Comment · Reviewer_PsXw · 2021-09-29
> **Reviewer's response**
>
> The original version of the paper was poorly represented while the revised version dramatically changes, which address many of my concerns. I'm willing to raise the score.

---

### Official Review · Reviewer_sUrc · 2021-09-20
**A large collection of datasets and useful API, but falls short on task-specific details and benchmarking**

**Rating:** 7
**Confidence:** 3

**Strengths:**

- This fills a need, supplementing widely used representation learning benchmarks with datasets more suited to generation and transformation. This is largely due to the lack of existing datasets with large numbers of graphs (as opposed to individually large graphs)
- The number and variety of datasets is impressive, and provides great opportunities for machine learning research across many fields that use graph-structured data
- The Python API provides a very useful set of processing and evaluation tools for graph generation and transformation, which should enhance accessibility and reproducibility

**Weaknesses:**

- Motivation for including each dataset is not clear in many cases. For example, statements like "it is attractive for deep generative models to model protein structures" should be backed up by examples or references. Some datasets have even less motivation stated.
- Detail on the datasets is lacking. Recognizing that is an inherent disadvantage of including such a large number of datasets with limited space, it nonetheless makes it difficult to understand the structure of each dataset and what types of tasks one would use each for. The descriptions are also inconsistent in the main text -- for example, sometimes the number of graphs and graph sizes are discussed and sometimes they are not. The paper would benefit from a clear parallel structure across task descriptions.
- Benchmarking is only performed on one task (molecule generation). There is no benchmarking on any graph transformation datasets. The authors' argument that they only benchmark one task because graph generation is domain specific is at odds with the goal of creating a unified benchmark with standardized evaluations. Without defining domain-specific evaluation criteria for each task, it is much harder for researchers to compare models consistently, harming the utility of the benchmark as a whole.
- Data format is not described, nor are the hyperparameters or experiment settings for the benchmark.

**Additional Feedback:**

This is an ambitious project and impressive array of datasets. I also think the Python toolkit will be useful for practitioners. However, the lack of detail in dataset construction and processing, combined with the lack of systematic benchmarking or even specification of evaluation criteria for each task, limit the impact of the work as currently presented.

**Clarity:**

The structure of the paper is clear, but clarity is hindered by the lack of detail due to the large number of tasks (particularly in Section 4). The description of graph generation could also be clearer.

**Correctness:**

There is little detail of dataset processing and construction for each task, and each domain has specific requirements, making it difficult to assess the quality of the datasets from the paper.

**Documentation:**

Yes

**Relation To Prior Work:**

Yes

**Summary And Contributions:**

This paper presents GraphGT, a large collection of 36 datasets for graph generation and graph transformation tasks, filling a need which is unmet by existing benchmarks for graph representation learning. These datasets span many fields, from biology to electrical engineering. In addition to the datasets, there is a Python package which provides a set of dataloaders and evaluation functions for each dataset and task. These datasets and tools provide a useful resource to researchers looking for graph datasets across many disparate fields.

---

> ### Author Response · Authors · 2021-09-27
> **Author's response (1/2)**
>
> We appreciate your valuable comments and feedback that help us improve the paper!
>
> Review #1: Motivation for including each dataset is not clear in many cases. For example, statements like "it is attractive for deep generative models to model protein structures" should be backed up by examples or references. Some datasets have even less motivation stated.
>
> Response: Thanks for your comments.
> * We have completely re-written the dataset details part (section 4.2, Line #167-303) with an organized format: motivation, task, data construction.
> * We elucidated motivations, added concrete examples, and related references for all domains covered by GraphGT.
> * We also provided the general procedures on the efforts we devoted when constructing the dataset, with even further details in appendix C (Line #783-1195).
>
> Review #2: Detail on the datasets is lacking. Recognizing that is an inherent disadvantage of including such a large number of datasets with limited space, it nonetheless makes it difficult to understand the structure of each dataset and what types of tasks one would use each for. The descriptions are also inconsistent in the main text -- for example, sometimes the number of graphs and graph sizes are discussed and sometimes they are not. The paper would benefit from a clear parallel structure across task descriptions.
>
> Response: Thanks for your comments.
> * We have provided the structure of each dataset and what types of tasks one would use each for in section 4.2 (Line #167-303)
> * We have made the descriptions consistent with a clear parallel structure, by aligning descriptions for each dataset in Section 4.2 (Line #167-303) into “motivation, tasks, data construction”.
> * We have provided a table, namely Table 1, that discusses the statistics of all the graphs to make the discussion consistent.
> * We have moved the detailed information from section 4.2 in the paper to the appendix C (Line #783-1195) in a more extensive and consistent format including dataset statistics, available features, as well as collection/processing details.
>
> Review #3: Benchmarking is only performed on one task (molecule generation). There is no benchmarking on any graph transformation datasets. The authors' argument that they only benchmark one task because graph generation is domain specific is at odds with the goal of creating a unified benchmark with standardized evaluations. Without defining domain-specific evaluation criteria for each task, it is much harder for researchers to compare models consistently, harming the utility of the benchmark as a whole.
>
> Response:  Thanks for your comments.
> * We have added new graph generation benchmark results and graph transformation benchmark results and analyses for 16 graph generation datasets and 17 graph transformation datasets with popular graph generation models in section 5.1.2 (Line #342-360) and section 5.2.2 (Line #386-408).
> * The details of the experiment setting can be found in Appendix D (#Line 1219-1277) including baseline models, hyper-parameters, experiment set-ups.
> * We also added the code to run the models on our GitHub Repo (https://github.com/yuanqidu/GraphGT).
> * We have added three more metrics to evaluate graph transformation models. GraphGT includes a list of 22 evaluation metrics. For domain-specific graph generation, we provide the evaluation metrics (e.g. validity) for existing well-developed domains. We have been regularly updating GraphGT to include more and more domains, datasets, evaluation metrics!
>
>
> Review #4: Data format is not described, nor are the hyperparameters or experiment settings for the benchmark.
>
> Response: Thanks for your comments.
> * We have added the descriptions of our unified data format in Appendix B (#Line 771-782).
> * We have additionally provided a detailed format description for each dataset in Appendix C (Line #783-1195).
> * We have enumerated all details including experiment set-up, hyper-parameters in Appendix D (#Line 1219-1277).
>
> Review #5: There is little detail of dataset processing and construction for each task, and each domain has specific requirements, making it difficult to assess the quality of the datasets from the paper.
>
> Response: Thanks for your comments.
> * We have included more organized information about each dataset in section 4.2 (Line #167-303) including motivations, tasks, and dataset construction.
> * We have included more detailed information in Appendix C including dataset statistics, available features, as well as collection/processing details (Line #783-1195).

---

> > ### Comment · Reviewer_sUrc · 2021-09-27
> > **Revision addresses most of my concerns**
> >
> > Thank you for the detailed response. The revisions address many of my concerns, and I am thus raising my score.

---

> ### Author Response · Authors · 2021-09-27
> **Author's response (2/2)**
>
> Review #6: The structure of the paper is clear, but clarity is hindered by the lack of detail due to the large number of tasks (particularly in Section 4). The description of graph generation could also be clearer.
>
> Response: Thanks for your comments.
> * We have rewritten Section 4.2 (Line #167-303) by aligning and re-organizing the content into “motivations”, “task descriptions”, and “dataset construction” for each dataset domain.
> * We have added more related work and refined graph generation and transformation background in sections 3.1 (Line #129-134) and 3.2 (Line #149-156).
> * We have included more detailed information in Appendix C including dataset statistics, available features, as well as collection/processing details (Line #783-1195).

---

### Author Response · Authors · 2021-09-27
**Thanks for the valuable feedback. We have significantly revised our paper**

We appreciate the valuable comments and feedback from all reviewers that prompt us to greatly improve the paper. We have made significant revisions to the text and have updated the PDF. While answering detailed questions of each review, we also list a summary of the updates:
* We have added graph generation benchmark results and graph transformation benchmark results and analyses for 16 graph generation datasets and 17 graph transformation datasets with popular graph generation models in section 5.1.2 (Line #342-360) and section 5.2.2 (Line #386-408), the experiment details can be found in Appendix D (Line #1196-1277) including baseline models, hyper-parameters, experiment set-ups. We also added the code to run the models on our GitHub Repo (https://github.com/yuanqidu/GraphGT).
* We have added more discussions about previous graph generation datasets, listed 5 main challenges that stifled the advances of the field, elaborated how GraphGT could help to tackle the challenges as well as enriched our motivations.  (In Section 1. Line #45-87)
* We have re-written our dataset description section 4.2 (Line #167-303) following an organized format with motivation, task, and data construction. Furthermore, we have moved all dataset details to Appendix C (Line #783-1195) with a unified format to include dataset statistics, available features, as well as collection/processing details.
We have added details regarding how many datasets we collected, re-purposed and reformatted as our contribution. (In Section 1 Line #73-78)
* We have added related works of graph generation and graph transformation. (In Section 3.1 and 3.2 Line #129-134, #149-156)
* We have added an illustrative figure that summarizes GraphGT. (Figure 1)
* We have added three more metrics to evaluate graph transformation models. (In Section 5.2 Line #368-370)
* We have added the unified dataset format for all datasets in GraphGT in Appendix B (Line #771-782).

---

### Author Response · Authors · 2021-09-29
**Any comments?**

Dear reviewers and AC,

Thanks a lot for your effort in reviewing this submission! We have revised and improved our paper significantly and tried our best to address any mentioned/potential concerns/problems during the rebuttal. Feel free to let us know if there is anything unclear or so. We are happy to clarify them.

Best Regards,
Authors

---

### Decision · Program_Chairs · 2021-10-09

**Decision:**

Accept

**Comment:**

All reviewers voted for acceptance. I recommend the authors to take into account the reviewers' comments and improve the paper for its camera-ready version.